# A Bayesian Multi-agent Multi-arm Bandit Framework for Optimal Decision Making in Dynamically Changing Environments

## Abstract

We introduce *DAMAS (Dynamic Adaptation through Multi-Agent Systems)*, a novel framework for decision-making in non-stationary environments characterized by varying reward distributions and dynamic constraints. Our framework integrates a multi-agent system with Multi-Armed Bandit (MAB) algorithms and Bayesian updates, enabling each agent to specialize in a particular environmental state. DAMAS continuously estimates the probability of being in each state using only reward observations, allowing rapid adaptation to changing conditions without the need for explicit context features. Our evaluation of DAMAS included both synthetic environments and real-world web server workloads. Our results show that DAMAS outperforms state-of-the-art methods, reducing regret by around $40\%$ and achieving a higher probability of selecting the best action.

## 1 Introduction

In today's technology-driven world, adaptive decision-making systems play a crucial role in various domains, ranging from industrial automation (Scordino et al., 2020) to financial trading (Liu et al., 2022) and online services (Min-Allah et al., 2021; De Sanctis et al., 2020), including web servers. These systems must process data and respond quickly to ensure reliable performance. E.g., consider a web server with client request patterns that vary throughout the day, ranging from $1 - 5ms$ to $10 - 15ms$. Although normalizing response times to a fixed range (for example, $[0, 1]$) simplifies decision-making, it becomes ineffective when distributions vary. Real-time workloads in web servers are dynamic and unpredictable, posing challenges in maintaining optimal performance. Static configurations often fail to adapt to changing demand patterns, leading to inefficiencies and service degradation (Araújo & Holmes, 2021).

The exploration-exploitation trade-off is crucial in managing systems. Balancing the exploitation of known configurations and the exploration of new ones with uncertain outcomes is fundamental in sequential decision-making and reinforcement learning (Hu & Xu, 2020; Shi & Xu, 2019; Efroni et al., 2020). Bayesian models, which account for uncertainty, often offer reliable solutions in dynamic environments (Galioto & Gorodetsky, 2020).

Many existing solutions for non-stationary multi-armed bandit problems assume the availability of contextual information or require engineered features that describe the environment's state (Zheng et al., 2023). However, in real-world applications, such contextual signals are often unavailable, unreliable, or costly to obtain, limiting the practical utility of these methods (Ghoorchian et al., 2024).

Hence, it is essential to develop multi-armed bandit algorithms for non-stationary environments. Many relevant algorithms for non-stationary environments such as Sliding-Window UCB (Garivier & Moulines, 2008), Sliding-Window TS (Trovo et al., 2020), and the approaches proposed by Cavenaghi et al. (2021), assume uniform rewards between 0 and 1, or -1 and 1. However, when rewards fall outside these ranges or when the range itself changes over time, these algorithms struggle to maintain optimal performance.

Hence, we propose DAMAS (Dynamic Adaptation through Multi-Agent Systems with Multiple Q-values), a novel framework for adaptation in non-stationary environments in multi-armed bandits.

Unlike prior approaches, DAMAS operates solely on observed rewards without relying on external context or engineered features. DAMAS combines the multi-armed bandits (MAB) algorithms with Bayesian updates, leveraging a multi-agent system consisting of specialized agents for different workload scenarios. Our approach seamlessly integrates the chosen MAB algorithm's ability to balance exploration and exploitation with Bayesian updates for environmental state estimation.

The key contributions of this paper are as follows: (i) We propose DAMAS, a framework for dynamic environments that integrates MAB algorithms with Bayesian updates to estimate the current environment and update the Q-values based on the current uncertainty. (ii) We show that DAMAS effectively handles varying reward ranges by employing multiple Q-values and specialized agents for different environmental states. (iii) We introduce Bayesian Optimization for hyper-parameter adjustment in DAMAS, which accounts for the uncertainty in agent performance across different environments. We tested DAMAS in synthetic and real-world web server environments, showing a 40% reduction in regret and a high optimal action selection.

## 2 RELATED WORKS

The challenge of optimal decision-making in dynamically changing environments has been a significant focus in the development of adaptive algorithms, particularly in the context of adaptive systems and multi-armed bandit problems. Studies have demonstrated adaptive decision-making using various techniques, such as chaotic semiconductor lasers for dynamically changing reward environments (Oda et al., 2022) and distributed consensus algorithms for multi-agent multi-armed bandits in dynamic settings (Cheng & Maghsudi, 2024). However, existing solutions often face difficulties in managing the combination of dynamic constraints, fluctuating reward distributions, and the need for rapid adaptation in various domains (De Curtò et al., 2023; Balef & Maghsudi, 2023).

**Multi-Armed Bandit Algorithms for Non-Stationary Environments:** Kaufmann et al. (2012) presented the Bayesian Upper Confidence Bound (UCB) algorithm, a variant of UCB1, which uses Bayesian inference to update beliefs about reward probabilities for action selection. DAMAS instead uses Bayesian inference to continuously update the probabilities of being in different environmental states. Cavenaghi et al. (2021) proposed a concept drift-aware algorithm for non-stationary multi-armed bandits, demonstrating improved performance in detecting and adapting to changes. Trovo et al. (2020) introduced a sliding-window Thompson sampling approach for non-stationary settings, offering a more nuanced adaptation mechanism. Similarly, Jia et al. (2023) introduced smooth non-stationary bandits, leveraging continuity assumptions to achieve better regret bounds. Chen et al. (2023) address non-stationary MABs by modeling temporal dependencies using an auto-regressive structure, introducing mechanisms to balance exploration-exploitation and reset outdated information, and achieve near-optimal regret bounds in dynamic environments. However, although these approaches show promise in handling changing environments, they often assume normalized reward ranges (e.g., $[0, 1]$), which may significantly limit their applicability in real-world scenarios where reward scales can vary dramatically across different environments.

Moreover, many existing bandit algorithms assume access to contextual information or rely on engineered features to characterize the environment's state (Zheng et al., 2023). In practice, however, such context signals may be noisy, incomplete, or costly to obtain (Bouneffouf et al., 2017; Ghoorchian et al., 2024). In contrast, DAMAS operates entirely on observed rewards without requiring any side information or context features, making it a more general solution suitable for any real-world adaptive systems.

**Bayesian Optimization for Adaptive Learning:** Bayesian optimization has emerged as a powerful method of adaptive learning, particularly in scenarios where uncertainty plays an important role (Cheng et al., 2022). This approach leverages probabilistic models to guide the optimization process, making it suitable for applications where the cost of function evaluations is high and data is scarce (Wei et al., 2021; Hong et al., 2024). The study of Li (2023) introduces a suite of uncertainty quantification methods for Bayesian deep learning, demonstrating the application of Bayesian principles to achieve robust and adaptive models. In Wei et al. (2021), the authors introduce Collaborative and Adaptive Bayesian Optimization (CABO), which combines Bayesian probabilistic optimization and integration. The proposed method focuses on handling mixed uncertainties in computational mechanics using an active learning method. Krishnamoorthy & Paulson (2023) proposed a multi-agent Bayesian optimization framework that uses the alternating direction method of multipliers (ADMM)

to solve black-box optimization problems over a multi-agent network. This approach addresses the coupling between subsystems without data sharing. DAMAS builds upon these Bayesian ideas for the continuous adaptation of exploration parameters and updating all agents simultaneously based on environmental probabilities.

**Bayesian Methods for Adaptive Systems:** Bayesian estimation plays a crucial role in predicting and adapting to changes in workload in adaptive systems. Using probabilistic models, Bayesian methods provide a robust framework to manage uncertainties and improve system performance (Galioto & Gorodetsky, 2020; do Carmo Alves et al., 2024). This adaptive capability is essential in environments where conditions can change rapidly and unpredictably. Xu et al. (2021) introduced a Bayesian inference framework to estimate the topology and state of a power distribution system. This adaptive method uses limited measurements to efficiently recover the system's Bayesian posterior distributions, enhancing the system's robustness and reliability in real-time. applications.

**Exploration-Exploitation Trade-offs:** In adaptive systems, the ability to dynamically adjust between exploration and exploitation is vital to maintaining optimal performance under changing conditions. Efroni et al. (2020) studied exploration-exploitation in constrained MDPs, providing theoretical insights into optimal strategies. Hu & Xu (2020) proposed an adaptive exploration strategy using multi-attribute decision-making for reinforcement learning. However, these works did not specifically address the challenges of adaptive systems with varying ranges.

## 3 METHODOLOGY

**Problem Formulation:** The dynamic environment can be modeled as a system that faces changing conditions (e.g., a web server that experiences varying workloads), represented by a set of environments $\mathbf{E} = \{e_1, e_2, e_3, ..., e_n\}$. Each environment $e_i \in \mathbf{E}$ is characterized by a set of means $\mu_i(a)$ and standard deviations $\sigma_i(a)$ for the rewards associated with different actions $a \in \mathbf{A}$, where $\mathbf{A}$ is the set of possible actions. The current environment transitions between these different environments in $\mathbf{E}$ periodically, and the transition dynamics are unknown to the agents. Importantly, the $\mu_i$ and $\sigma_i$ of each environment can be defined in any arbitrary way across time, actions, or both. This generality allows our framework to capture the different types of non-stationary bandit problems, including: (i) abrupt changes, (ii) gradual changes (Komiyama et al., 2024), (iii) recurring/seasonal changes (Keerthika & Saravanan, 2020), and (iv) random changes (Cavenaghi et al., 2021).

To address the dynamic environment, we propose a multi-agent system consisting of a set of agents $\mathbf{\Phi} = \{\phi_1, \phi_2, \phi_3, ..., \phi_n\}$, where each $\phi_i$ corresponds to the environment $e_i$. Each agent $\phi \in \mathbf{\Phi}$ is responsible for selecting actions and updating its Q-values $Q(\phi, a)$ based on the observed rewards $r$ and environmental probabilities $P(e_i)$.

In this context, we adapt Q-value-based Multi-Armed Bandit (MAB) algorithms for our multi-agent setting to select actions. For example, we adapt the upper confidence bound (UCB1) algorithm to work with multiple agents. The adapted version of the UCB1 equation is: $a^* = \underset{a}{\mathrm{argmax}} \left( Q(\phi, a) + c\sqrt{\frac{2 \log(t)}{N(\phi, a)}} \right)$, where $a^*$ is the selected action, $Q(\phi, a)$ is the estimated Q-value for action $a$ by agent $\phi$, $t$ is the total number of trials, $N(\phi, a)$ is the number of times action $a$ has been selected by agent $\phi$, and the exploration constant $c$ controls the degree of exploration.

As shown in Algorithm 1, each agent $\phi$ is an instance of the chosen multi-armed bandit algorithm (e.g., UCB1), with separate Q-values $Q(\phi, a)$. From the perspective of our multi-agent system, we initialize the probability of being in each environment $e_i$ with an initial prior distribution, i.e., $P(e_i) = \frac{1}{|\mathbf{E}|} \quad \forall e_i \in \mathbf{E}$.

At each time step $t$, all agents select their preferred actions based on their respective algorithms. The final action $a_t$ is then sampled based on the probabilities of the current environment $P(e_i) := P(e_i|r_t)$. The sampled environment determines which agent's perspective is used for action selection. However, note that Q-value updates are performed for all agents, regardless of which agent determined the action taken by the system. That is, the system takes the action chosen by the agent $\phi_i$, which is sampled with probability $P(e_i)$.

**Q-value Updates:** After observing the reward $r_t$ for action $a_t$, we update for each agent $\phi \in \Phi$: $S(\phi, a_t) \leftarrow S(\phi, a_t) + P(e_\phi) \cdot r_t$, $N(\phi, a_t) \leftarrow N(\phi, a_t) + P(e_\phi)$, $Q(\phi, a_t) \leftarrow \frac{S(\phi, a_t)}{N(\phi, a_t)}$. This update

method allows for incremental adjustments to the Q-values based on the observed rewards and the probability of being in each environment. By weighting both the reward and the count increment by $P(e_\phi)$, we are estimating the expected Q-value and the expected number of trials.

**Environment Probability Updates:** The probabilities of being in each environment $e_i$ are updated using Bayesian updates. Given the observed reward $r_t$ and the current Q-values $Q(\phi, a_t)$ of the agents, we calculate the likelihood of observing $r_t$ in each environment $e_i$ using Gaussian probability density functions: $P(r_t|e_i) = \mathcal{N}(r_t; \tilde{\mu}_i(a_t), \tilde{\sigma}_i(a_t)^2)$, where $a_t$ is the action taken by the system corresponding to the sampled environment, $\tilde{\mu}$ and $\tilde{\sigma}$ represent the estimated mean and standard deviation. Note that action $a_t$ is not necessarily the action preferred by agent $\phi_i$ at iteration $t$, but corresponds to the action chosen by one of the agents through the sampling process explained previously.

These likelihoods are then combined with the current environment probabilities $P(e_i)$ (which serve as priors for this update) using Bayes' theorem to obtain the posterior probabilities of being in each environment at iteration $t$, given the observed reward $r_t$: $P(e_i \mid r_t) \propto P(r_t \mid \tilde{e}_i) \cdot P(e_i)$

Then the posterior probabilities are normalized to ensure that they sum to 1 as: $P(e_i|r_t) = \frac{\tilde{P}(e_i|r_t)}{\sum_j \tilde{P}(e_j|r_t)}$.

The updated probabilities $P(e_i|r_t)$ become the new priors for the next iteration, for each environment. These steps are repeated iteratively, at each iteration, the Q-values of the agents, the probability estimates of the environment and the estimates $\tilde{\mu}$ and $\tilde{\sigma}$ are refined.

Before the main process, each agent undergoes a pre-training phase to initialize its knowledge and decision-making capabilities. This process is crucial for establishing a baseline understanding of different workload scenarios. In this phase, each agent $\phi$ is trained on a static environment for each $e_i$. This pre-training allows the agents to initialize their likelihood functions $P(r_t|e_i)$ and gain experience in their respective scenarios. Here, agents establish initial estimates $\tilde{\mu}_i(a)$ and $\tilde{\sigma}_i(a)$ for the reward distributions of each action in each environment. Moreover, while pre-training focuses on specific environments, the DAMAS framework is designed to handle a mixture of environments and scenarios not explicitly covered in the pre-training phase. During actual execution, the system may encounter environments that differ from those in the initial set $\mathbf{E}$, or even face a different number of environments than initially anticipated. In these cases, DAMAS leverages its probability estimation mechanism to approximate the current environment as a mixture of pre-learned environments.

Following pre-training, the main simulation forms the core of our dynamic adaptation process and iterates over a specified number of time steps $T$. In each time step $t \in 1, 2, \ldots, T$, the multi-agent system performs the following sequence of operations, (i) agent sampling, (ii) action selection, (iii) action execution and reward observation, (iv) Q-value updates, (v) environment probability updates. This iterative process enables the system to continuously learn and adapt to the dynamic environmental change.

**Optimization of $c$ Parameters Using Bayesian Optimization (BO):** The optimization of the configuration parameters for each agent's MAB algorithm is performed using Bayesian optimization. While this approach can be applied to any parameter of the chosen MAB algorithm, we will use the exploration parameter $c$ of UCB1 as an illustrative example. BO consists of two main components: (i) a Gaussian Process (GP) model of the objective function $f$, which captures the underlying reward dynamics; and (ii) an acquisition function, which guides the selection of the next set of $c$-values to evaluate (Shahriari et al., 2015; Frazier, 2018).

*A Gaussian Process (GP)* model is fitted to the observed response times. Let $\mathbf{c} = \{c_1, c_2, \ldots, c_m\}$ be the set of configuration parameters, and $\mathbf{r} = \{r_1, r_2, \ldots, r_m\}$ be the corresponding reward. The GP model assumes that the rewards are drawn from a Gaussian distribution: $\mathbf{r} \sim \mathcal{N}(\mu, \mathbf{K} + \sigma_n^2 \mathbf{I})$, where $\mu$ is the mean vector, $\mathbf{K}$ is the covariance matrix defined by the Radial Basis Function (RBF) kernel, $\sigma_n^2$ is the noise variance, and $\mathbf{I}$ is the identity matrix. Given a test point $c^*$, the posterior mean $\mu(c^*)$ and covariance $\sigma^2(c^*)$ are computed as:

$\mu(c^*) = \mathbf{k}_*^T (\mathbf{K} + \sigma_n^2 \mathbf{I})^{-1} \mathbf{r}$, $\sigma^2(c^*) = k(c^*, c^*) - \mathbf{k}_*^T (\mathbf{K} + \sigma_n^2 \mathbf{I})^{-1} \mathbf{k}_*$, where $\mathbf{k}_* = [k(c^*, c_1), k(c^*, c_2), \ldots, k(c^*, c_n)]^T$ is the covariance vector between the test point and the training points. The Lower Confidence Bound (LCB) is used as the acquisition function to select the next configuration parameter, which explores more diverse input space regions than Expected Im-

provement (EI) (Figure 17 in the Appendix): $\text{LCB}(c) = \mu(c) - \kappa\sigma(c)$, where $\kappa$ is a parameter that balances exploration and exploitation. The $c$-value that minimizes the LCB is selected as $c_{\text{new}} = \arg\min_c \text{LCB}(c)$.

**Updating All Agents:** To update all agents simultaneously, the posterior mean $\mu(c)$ is divided by the total sum of probabilities $\sum_t P(e_i \mid r_t)$ for the specific environment and then multiplied by the total number of observations $N_i$ for that environment: $\mu_{\text{updated}}(c) = \frac{\mu(c)}{\sum_t P(e_i|r_t)} \times N_i$.

This ensures that the optimization process accounts for the probability and the number of observations for each environment, which may lead to a more robust and adaptive configuration update.

By incorporating Bayesian optimization and updating all agents simultaneously, the system can effectively learn and adapt the configuration parameters $c$ to minimize the cost, thus improving the overall performance of the system in dynamic environments (*see Appendix A for DAMAS in action and a pseudo-code*).

## 4 THEORETICAL ANALYSIS

**Convergence of Environment Probabilities:** In our paper we assume dynamically changing environments. In the following theorem we show that if the agent is in a fixed but unknown environment, it will converge to the same regret as UCB1.

**Theorem 4.1.** *Let $e^*$ be the true environment. If $r_t \sim P(r_t|e^*)$ and $T \to \infty$, the following holds: $P(e^*|r_t) \to 1$ and $P(e_j|r_t) \to 0, \forall j \neq i$. The DAMAS framework reduces to a standard UCB1 algorithm in environment $e^*$, achieving regret $R(T) = O(\log T)$ as $T \to \infty$. (**see Appendix C for the full proof**).*

**Convergence in Dynamically Changing Environments:** If the agent operates in a single stationary environment $e_i$ for a sufficiently long time, it will converge to behavior equivalent to the UCB1 algorithm for that environment. When the environment transitions to another environment $e_{i+1}$, if the agent remains in this new environment for long enough, it will again converge to UCB1 behavior for $e_{i+1}$.

To formalize this, we use the concept of ordinal numbers, which allows us to represent infinite sequences of infinite phases. This framework models the timeline where the agent successively adapts to each environment, achieving phase-wise convergence.

*Ordinal numbers:* extend natural numbers to represent ordered sequences, including infinite ones. For example: $\Omega = \{0, 1, 2, \ldots, \omega, \omega + 1, \ldots\}$ represents a timeline where $\omega$ is a countably infinite phase, and $\omega + 1$ marks the beginning of the next phase. In our setting, each phase $\omega_i$ corresponds to an infinite number of iterations in environment $e_i$.

***Setting and Assumptions***: (i) Consider a multi-armed bandit system with a finite set of environments $E = \{e_1, e_2, \ldots, e_k\}$, with $|E| = k$, where each $e_i$ represents a stationary environment. (ii) The agent operates in a single environment $e^* \in E$ for an infinite number of iterations, denoted $\omega_i$. (iii) After $\omega_i$ iterations in $e_i$, the environment transitions to $e_{i+1}$. (iv) This process repeats indefinitely, creating an *ordinal timeline* $\Omega$, which is defined as: $\Omega = \omega_i$. The ordinal numbers $\Omega$ represent a hierarchy of infinite iterations. (v) The reward distribution $r_t \sim P(r_t|e_i)$ remains stationary within each phase $e_i$.

**Theorem 4.2.** *Let $E = \{e_1, e_2, \ldots, e_k\}$ be the set of environments. Assume the agent stays in each environment $e_i$ for $T_i \to \infty = \omega_i$ iterations. As $T \to \infty$, the following holds:*

*Within each environment $e_i$: $P(e_i|r_t) \to 1$, and $P(e_j|r_t) \to 0$, $\forall j \neq i$. Across all environments $E$: The cumulative regret over $\Omega$ is still: $R(T) = O(\log T)$. (**see Appendix D for the full proof**).*

## 5 EXPERIMENTAL RESULTS

We evaluate the effectiveness of our proposed DAMAS framework using simulations and a real-world web server environment. We begin by presenting the results from our simulation experiments to assess the framework's performance under various conditions. The framework's goal is to minimize response times, which are treated as costs, thereby maximizing cumulative reward across mul-

tiple environments. This analysis investigates the multi-agent system's decision-making prowess and adaptability across varying workload scenarios. For each action $a$ in the environment $e$, the associated cost (response time) is modeled by a normal distribution $N \sim (\mu_{a_e}, \sigma_{a_e})$. The reward is inversely related to the cost, defined as the negative of the cost ($-$cost), implying that a lower cost yields a higher reward for agents. We evaluated DAMAS effectiveness through several metrics:

(i) **Probability of Selecting the Best Action** ($P_{best}$): $P_{best}(\pi) = \frac{1}{T} \sum_{t=1}^{T} I(a_t^* = a_t)$, where $a_t^*$ is the optimal action at time $t$, and $I(\cdot)$ is the indicator function. (ii) **Mean Cumulative Sampled Regret (MCSR)**: $\text{MCSR}(\pi) = \frac{1}{T} \sum_{t=1}^{T} (r_t(a_t^*) - r_t(a_t))$, where $r_t(a_t^*)$ is the sampled reward for the optimal action at time $t$, and $r_t(a_t)$ is the sampled reward for the action chosen by the algorithm $\pi$. For simplicity and consistency, we refer to this metric as **Mean Regret (MR)** throughout the rest of the paper. (iii) **Mean Response Time (MRT)** for a given policy $\pi$: $\text{MRT}(\pi) = \frac{1}{T} \sum_{t=1}^{T} r_t$, where $r_t$ represents the response time observed at time $t$ and $T$ is the total number of time steps.

We benchmark our approach DAMAS against other MAB algorithms, including (i) UCB1 (Auer et al., 2002), (ii) Bayesian UCB (Kaufmann et al., 2012), (iii) Sliding-Window UCB (Garivier & Moulines, 2008) with $windowsize = 100$ that captures the most recent actions and rewards observed, and $c = 1$ which represents the exploration constant, (iv) Discounted UCB (Garivier & Moulines, 2008) with $\gamma = 0.99$: discount factor with $0 < \gamma \leq 1$, $c = 1$, (v) Mean f-dsw TS (with mean as an aggregation function) (Cavenaghi et al., 2021) with $windowsize = 100$ and $\gamma = 0.9$, and (vi) Sliding-Window TS (Trovo et al., 2020) with $windowsize = 100$.

**Comparative Analysis: DAMAS Variants vs. Baselines:** To assess the effectiveness of different components within the DAMAS framework, we compare multiple DAMAS-enhanced configurations directly against their original MAB algorithms. The DAMAS variants include: (i) DAMAS-UCB a multi-agent system that uses UCB1 with fixed hyper-parameters ($c$), (ii) BO-DAMAS-UCB extends DAMAS-UCB by incorporating Bayesian optimisation to adapt hyper-parameters dynamically, and (iii) DAMAS-SW-UCB, DAMAS-Dis-UCB, DAMAS-SW-TS, and DAMAS-Mean d-sw TS these configurations apply the DAMAS framework to their underlying algorithms. Moreover, we include comparisons between the single-agent BO-UCB (which uses Bayesian optimisation to tune hyper-parameters dynamically) and its multi-agent extension BO-DAMAS-UCB.

**Workload Characteristics Across Multiple Environments:** In our multi-agent system, we model three environments to evaluate the DAMAS framework's adaptability across varying scales of response times and volatilities. The first environment simulates a web server, which features mean response times ($\mu$) from 0.2 to 0.7 seconds and standard deviations ($\sigma$) from 0.1 to 0.25, with an optimal action at $\mu = 0.2$ and and $\sigma = 0.03$. The second is a micro workload environment, with $\mu$ ranging from 0.02 to 0.09 seconds and $\sigma$ from 0.01 to 0.05, with an optimal action at $\mu = 0.015$ and and $\sigma = 0.02$. The third, representing high-load scenarios, has $\mu$ ranging from 1.5 to 2.5 seconds and $\sigma$ from 0.1 to 0.3, with an optimal action at $\mu = 1.0$ and and $\sigma = 0.4$.

To thoroughly assess the DAMAS framework, we test it under three dynamic scenarios: (i) abrupt changes, where the system transitions suddenly between environments; (ii) incremental changes, where shifts occur gradually over time; (iii) random changes, where environment switches occur stochastically according to a hazard function with probability 0.001. (iv) and mixed scenarios, where environments blend according to the parameters $\alpha$ and $\beta$, where $0 \leq \alpha, \beta \leq 1$. These parameters control the contribution of each environment to the new mixed scenario. In the mixed scenario, the new environment's $\mu_{\text{mixed}}$ and $\sigma_{\text{mixed}}$ are calculated as weighted combinations of the original environments as: $\mu_{\text{mixed}} = \alpha \mu_{e_1} + \beta \mu_{e_2} + (1 - \alpha - \beta) \mu_{e_3}$ and $\sigma_{\text{mixed}} = \alpha \sigma_{e_1} + \beta \sigma_{e_2} + (1 - \alpha - \beta) \sigma_{e_3}$.

Each agent is pre-trained for 100 iterations in a fixed environment and then evaluated on a different set, exposing them to unseen dynamics.

**Baselines study:** We will undertake a comprehensive analysis of the DAMAS strategy and its variant in contrast to several baseline methods. As illustrated in Figure 1a for the sudden changes scenario, BO-DAMAS-UCB consistently shows the lowest mean response time in all the numbers of actions and high $P_{best}$ as shown in Figure 1b. Compared to the baseline methods UCB1 and B-UCB (Bayesian UCB), BO-DAMAS-UCB demonstrates an approximately $40 - 45\%$ lower Mean Regret (MR) , as shown in Figure 2a. For the same figure, other MAB algorithms, particularly DAMAS-Mean d-sw TS and DAMAS-SW-TS, outperform BO-DAMAS-UCB for a low number of

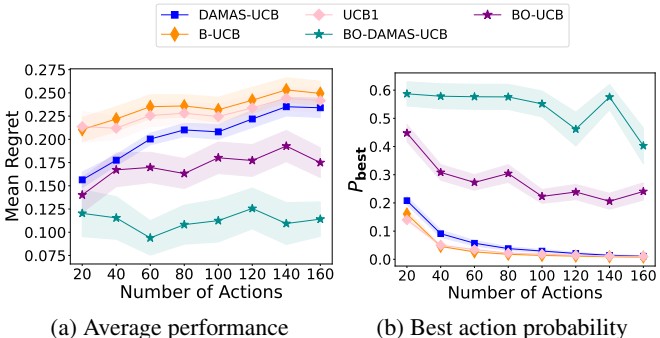

(a) Average performance  (b) Best action probability

Figure 1: Comparing best-performing approach across multiple actions for Sudden Change Scenario. UCB1, B-UCB, DAMAS-UCB, BO-UCB, and BO-DAMAS-UCB.

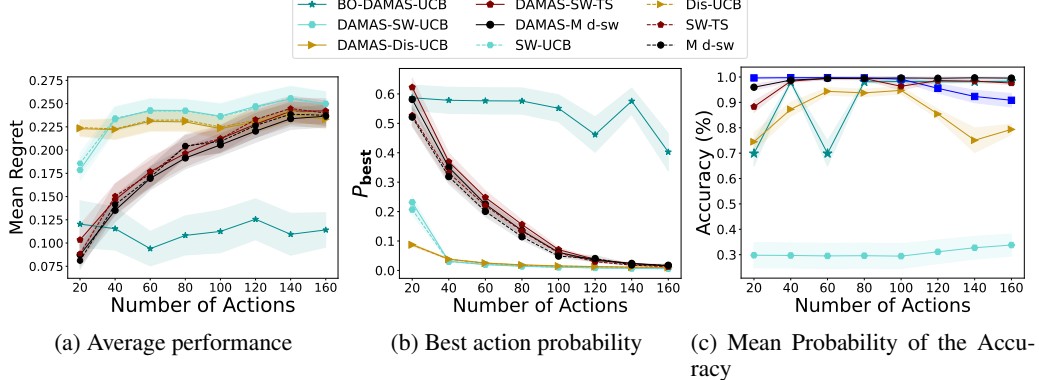

(a) Average performance  (b) Best action probability  (c) Mean Probability of the Accuracy

Figure 2: Comparing best-performing approach across multiple actions for Sudden Change Scenario. UCB1, B-UCB, DAMAS-UCB (multi-agent with Bayesian estimation and a fixed $c = 1$), BO-UCB, and BO-DAMAS-UCB.

actions, while BO-DAMAS-UCB showed more stability as the number of actions increased. The $P_{best}$ for BO-DAMAS-UCB showed five times higher than the next best algorithms for $a = 160$.

For gradual changes, as depicted in Figure 3 BO-DAMAS-UCB consistently outperforms other MAB algorithms, achieving the highest $P_{best}$ (around $40-50\%$) for higher action counts ($a = 160$). BO-UCB also shows competitive performance as the number of actions increases.

In the same incremental change scenario, as shown in Figure 3, initially the baselines Mean d-sw TS and SW-TS achieved lower MR for $a = 20$, but as the number of actions increased BO-DAMAS-UCB remained consistent and outperformed other baselines, achieving an average $\approx 25\%$ lower MR. Discounted UCB is the worst in these scenarios, particularly in the $P_{best}$. When applied to different MAB algorithms, DAMAS shows varying degrees of improvement. The application of DAMAS to Thompson Sampling particularly DAMAS-Mean d-sw TS, shows improvement over the original Mean d-sw TS with a high $P_{best}$.

In the mixed scenario ($\alpha = 0.3, \beta = 0.4$), BO-DAMAS-UCB maintained a higher performance with around $50\%$ lower MR compared to the UCB1 and B-UCB methods and DAMAS-UCB across all actions with 6 times higher in $P_{best}$ (Figure 4a, 4b). For other MAB algorithms, as shown in Figure 4c, 4d, similar conclusions can be drawn, with BO-DAMAS-UCB consistently outperforming other algorithms, achieving an average $\approx 40\%$ lower MR compared to the next best performer (SW-TS) in 160 actions. As a general remark, most MAB algorithms perform better in this scenario than abrupt and incremental changes.

Under randomly changing environments (Figure 5), BO-DAMAS-UCB and its variant achieve the lowest mean regret across most action sizes, maintaining values around $0.16$ at $a = 160$, compared to over $0.35$ for B-UCB and UCB1. Similarly, in Figures 5c and 5d, BO-DAMAS-UCB remains consistently, outperforming all competitors in both regret and optimality, particularly as the number of actions increases.

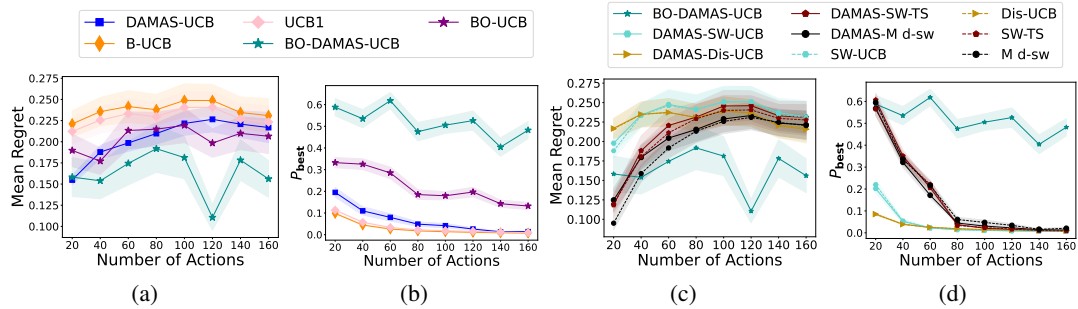

Figure 3: Performance comparison for Incremental Change scenario across varying action counts: (a,b) Baseline MAB vs. proposed approaches showing average regret and best action probability; (c,d) Extended comparison across multi-agent and single MAB algorithms. BO-DAMAS-UCB demonstrates better adaptation with increasing action space complexity.

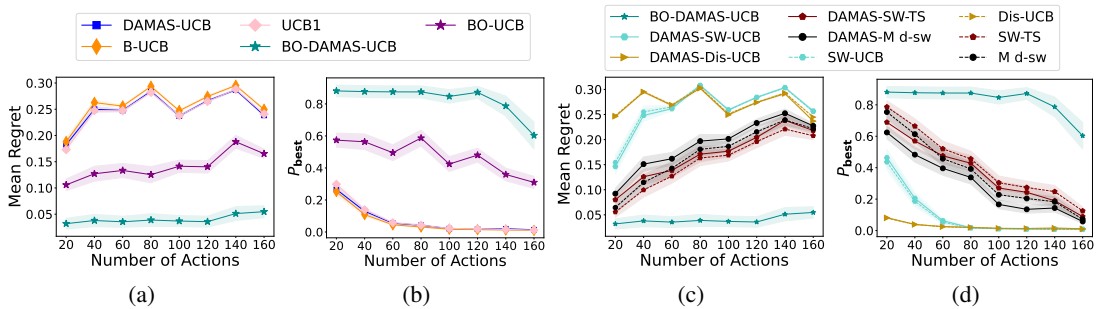

Figure 4: Comparing approaches across multiple actions for Mixed Scenario ($\alpha = 0.3$ and $\beta = 0.4$): (a,b) Ablation study showing BO-DAMAS-UCB outperforming other methods; (c,d) Comparison across MAB algorithms demonstrating BO-DAMAS-UCB's lower response times. Left plots show average performance/regret; right plots show best action probability.

**Scalability for the Number of Environments:** To further evaluate the scalability of DAMAS, we conducted additional experiments varying the number of underlying environments, $|\mathbf{E}|$, in the non-stationary bandit setting. We generated synthetic environments with $|E| \in \{3, 6, 9, 12, 15, 18, 21, 24\}$, where each environment $e_i$ defines distinct reward distributions for each of the $a = 80$ actions. The environment changes abruptly every 1000 time steps.

Figure 6 evaluates the scalability of DAMAS-based approaches as the number of underlying environments increases in a non-stationary bandit setting with a fixed action space of 80 actions (arms). In the environment scaling analysis (Figures 6a and 6b), BO-DAMAS-UCB consistently outperforms all other algorithms. At $|\mathbf{E}| = 24$, it achieves around $40\%$ lower MR than SW-TS and Mean d-sw TS. Interestingly, DAMAS-SW-UCB showed a lower MR around $25\%$ than SW-UCB. In the fixed setting of 9 environments in steps 9000 (Figures 6c and 6d), BO-DAMAS-UCB maintains the lowest MR and the highest $P_{best}$.

**Real-world web server:** For real-world web server environments (Porter et al., 2016) with diverse request types—including large image files unsuitable for caching or compression, text files amenable to compression, and cacheable image files, featuring sudden changes between distinct workload scenarios, each lasting 100 time steps. DAMAS-UCB and BO-DAMAS-UCB consistently maintain the lowest average response times in all workload scenarios (Figure 7). (*see Appendix B for more details on specific request patterns and file distributions*).

In conclusion, the effectiveness of DAMAS, particularly BO-DAMAS-UCB, arises from integrating a multi-agent approach and adaptive hyper-parameter learning. This combination enables the system to specialize in different environmental states while also scaling efficiently. Moreover, DAMAS not only outperforms existing approaches, but also generally improves base algorithms when integrated with them, though the degree of improvement varies depending on the base algorithm and environ-

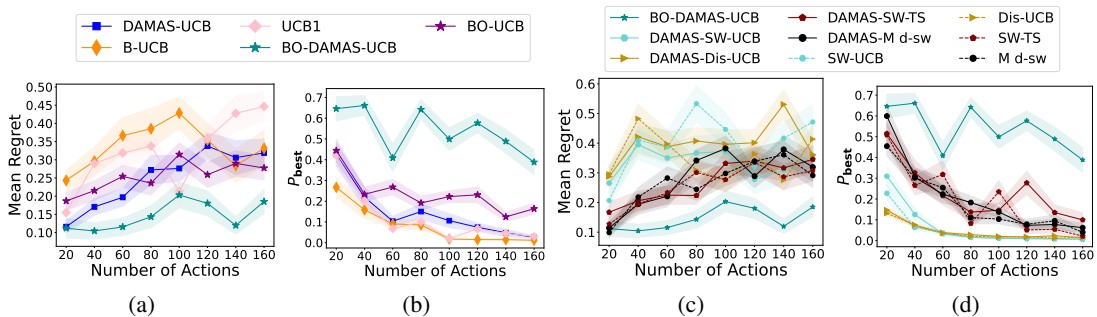

Figure 5: Performance comparison for Random Change scenario across varying action counts: (a,b) Baseline MAB vs. proposed approaches showing average regret and best action probability; (c,d) Extended comparison across multi-agent and single MAB algorithms.

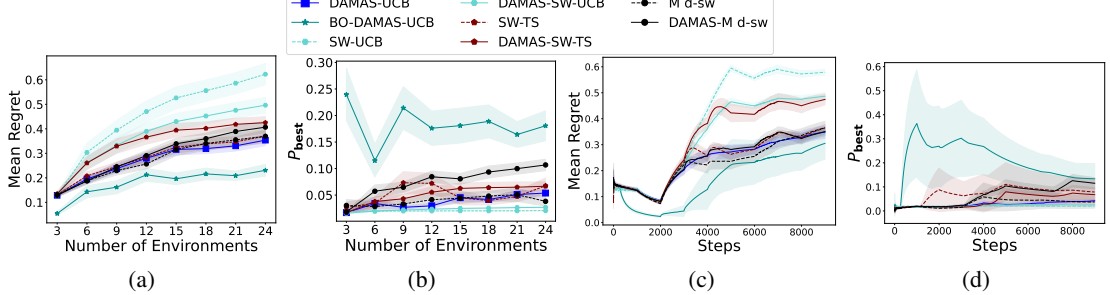

Figure 6: Comparing approaches across varying environments and over time: (a,b) Scaling behavior as number of environments increases from 3 to 24, showing BO-DAMAS-UCB maintains lowest mean regret (a) and highest probability of selecting the best action (b) across all environment counts; (c,d) Detailed analysis for a fixed setting of 9 environments showing algorithm performance over 9000 steps, with BO-DAMAS-UCB achieving better regret minimization (c) and maintaining higher best action selection probability (d).

mental conditions. Regarding computational efficiency and overhead, DAMAS maintains practical efficiency; for full details, see Appendix A.

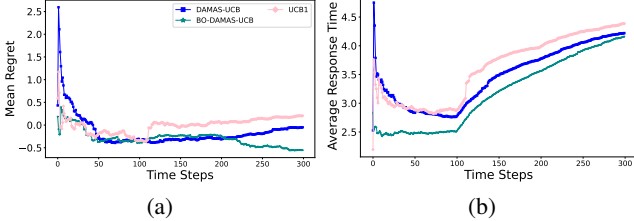

Figure 7: Average response time of Multi-Agent Approaches and baseline agent under three real workloads.

# 6 CONCLUSION

We introduced DAMAS, a novel framework for decision-making in non-stationary environments. It addresses the challenge of balancing exploration and exploitation in adaptive systems through a multi-agent approach integrated with MAB algorithms and Bayesian updates. Innovations include specialized agents for different environmental states, Bayesian optimization to tune parameters, and the ability to manage varying reward distributions. Experimental results in synthetic and real-world environments demonstrated DAMAS's ability to handle varying environmental change scenarios.

## REPRODUCIBILITY STATEMENT

The complete implementation code of DAMAS, all baseline algorithms, dataset generation scripts, and experimental pipelines are provided in the Supplementary Materials.

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

## A  ADDITIONAL EXPERIMENTS AND EXPLANATIONS

**Example: DAMAS in action - From Pre-training to Dynamic Adaptation:**  Before DAMAS begins operating in a dynamic environment, each agent undergoes a pre-training with fixed characteristics. After this phase, DAMAS is deployed in a dynamic environment where workloads fluctuate between $e_{low}$, $e_{micro}$, and $e_{high}$. Initially, we set equal probabilities for each $e$: $P(e_1) = P(e_2) = P(e_3) \approx 0.33$. At each time step, DAMAS first samples an agent according to the current environment probabilities. The selected agent then chooses an action $a_t$, which is executed in the environment. After executing $a_t$, a reward $r_t$ is observed. Now, DAMAS updates its belief about the current $e$. Calculate the probability of observing this $r_t$ after taking an action $a_t$ in each $e$ using Equation equation iv. Given our pre-training knowledge, and assuming that we observed a reward $r_t = 0.4$, we compute the likelihoods and observe the following for each $e_i$: $P(r_t \mid e_1) = 0.3$, $P(r_t \mid e_2) = 0.1$, and $P(r_t \mid e_3) = 0.01$. Using Bayes' theorem, we compute the *unnormalized* posteriors: $\tilde{P}(e_1 \mid r_t) = 0.3 \cdot 0.33 = 0.099$, $\tilde{P}(e_2 \mid r_t) = 0.1 \cdot 0.33 = 0.033$, and $\tilde{P}(e_3 \mid r_t) = 0.01 \cdot 0.33 = 0.0033$. To obtain the final probabilities, a normalization step is performed using the formula: $P(e_i \mid r_t) = \frac{\tilde{P}(e_i \mid r_t)}{\sum_j \tilde{P}(e_j \mid r_t)}$. For $e_1$, this yields $P(e_1 \mid r_t) = \frac{0.099}{0.1353} \approx 0.73$. This iterative process enables DAMAS to continually refine its understanding of the current state of the environment.

**Performance Analysis:**  When applied to a real web server with 38 different configurations, the results in Figure 8 provide valuable insight into the practical performance of BO-DAMAS-UCB. Figure 8b shows the accuracy of the algorithm in identifying the current workload environment. The first two workloads showed high accuracy (around 99%), followed by temporary drops during transitions, indicating that the system may struggle to identify the correct environment due to the similarity in response time. Furthermore, as shown in Figure 9, the DAMAS-based approaches consistently demonstrate improved responsiveness in three dynamic environments compared to the standard UCB1 algorithm. In the first environment (Figure 9a), BO-DAMAS-UCB achieved significantly lower response times—around 1.5 to 2.5 seconds—while UCB1 exhibits higher variance and a slower convergence to stability. During the transition to the second environment (Figure 9b), response times increase across all methods due to workload changes, but BO-DAMAS-UCB adapts more rapidly, maintaining a lead over DAMAS-UCB and a noticeable advantage over UCB1 throughout. In the final environment (Figure 9c), DAMAS-based methods continue to outperform, with BO-DAMAS-UCB maintaining the lowest response times (3.6–4.0 seconds).

Figures 10 and 11 compare DAMAS-based approaches with their single-agent counterparts under real-system workloads. In Figure 10, DAMAS-Mean dsw-TS exhibits higher mean regret and response time than the single-agent Mean dsw-TS. Similarly, in Figure 11, DAMAS-SW-TS shows a modest advantage in regret over SW-TS, particularly in later steps.

These results suggest that in this real-system setting, the multi-agent DAMAS variants may be disadvantaged by inaccurate environment inference as shown in Figure 12.

Figure 13 presents a comprehensive comparison of mean response times for varying numbers of actions under four dynamic environment scenarios: sudden change, incremental change, random change, and mixed change. Across all settings, BO-DAMAS-UCB consistently demonstrates the lowest or near-lowest response times. Similarly, as shown in Figure 14, BO-DAMAS-UCB in most scenarios achieves the lowest mean response times, demonstrating strong adaptability and scalability as the number of actions increases.

Figure 15 evaluates the performance of proposed and baseline MAB algorithms under an extreme random change setting, where the environment may switch at every time step (hazard probability = 1.0). Across both regret figures (Figures 15a and 15c), BO-DAMAS-UCB achieves the lowest mean regret across all action counts, demonstrating strong adaptability and resilience to frequent environmental changes. Importantly, in this experiment with three distinct environments, the probability of encountering the same environment is about 30%. As a result DAMAS leverages frequent belief updates and environment-weighted agents to quickly adapt and recover, using only reward feedback without needing contextual information.

As illustrated in Figure 16, we observe low response times of most number of environments, with BO-DAMAS-UCB and its variant (DAMAS-UCB) outperforming other MAB algorithms. These results highlight the scalability of the DAMAS framework in adapting to increasing environment complexity while maintaining efficient decision-making performance.

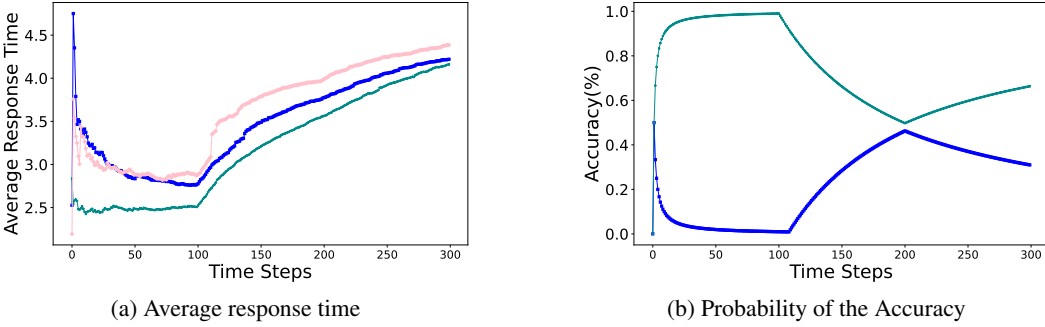

(a) Average response time            (b) Probability of the Accuracy

Figure 8: Average response time and probability of identifying the actual environment.

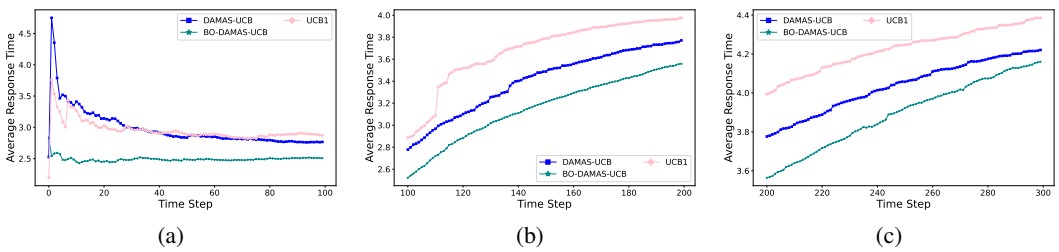

(a)                    (b)                    (c)

Figure 9: Average response time comparison across three different environments. DAMAS-UCB and BO-DAMAS-UCB demonstrate improved adaptability compared to the baseline UCB1 approach, particularly during workload transition periods.

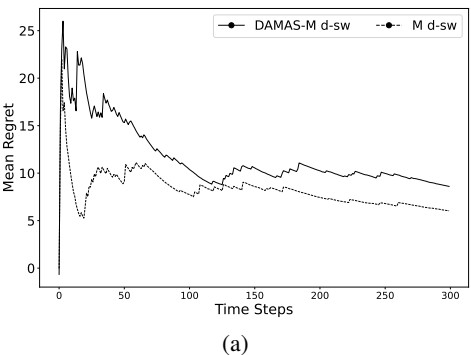 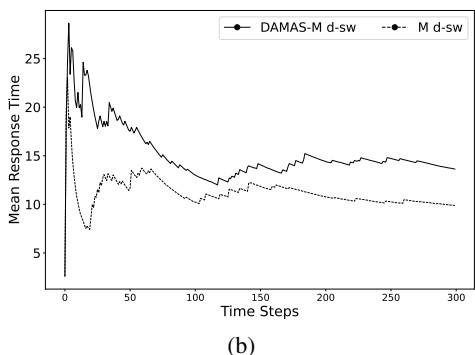

(a)                                                 (b)

Figure 10: Mean Regret and Average response time for DAMAS-Mean d-sw TS and Mean d-sw TS for actual real-system workloads.

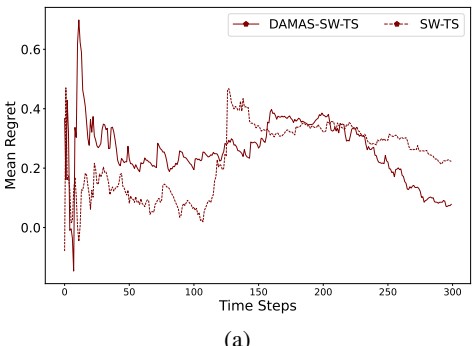 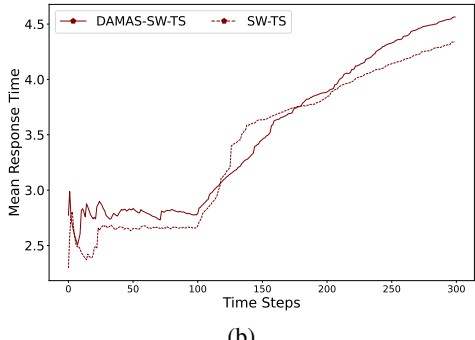

(a)                                                 (b)

Figure 11: Mean Regret and Average response time for DAMAS-SW TS and single SW TS for actual real-system workloads.

**Computational Efficiency and Overhead:** We conducted a 5-run benchmark comparing DAMAS to representative baselines over 6,000-step simulations across varying action counts. Despite its Bayesian multi-agent design, DAMAS maintains practical efficiency, averaging *39.3×* *slower than stationary UCB1*, but notably remains *1–3× faster than Sliding-Window UCB* for moderate-to-large action spaces ($A \geq 60$). Discounted UCB performs close to UCB1 (0.83×), but lacks DAMAS's adaptability. Full timing breakdowns are shown in the tables 1 and 2.

| Actions | DAMAS | Sliding-Win | Discounted | UCB1 |
|---------|-------|-------------|------------|------|
| 20 | 9.545 | 4.710 | 0.180 | 0.219 |
| 40 | 8.980 | 8.110 | 0.181 | 0.211 |
| 60 | 8.662 | 11.571 | 0.185 | 0.210 |
| 80 | 8.651 | 14.595 | 0.190 | 0.224 |
| 100 | 8.750 | 18.378 | 0.215 | 0.268 |
| 120 | 9.090 | 21.903 | 0.181 | 0.221 |
| 140 | 8.609 | 23.676 | 0.199 | 0.229 |
| 160 | 8.611 | 26.588 | 0.181 | 0.233 |

Table 1: Raw wall-clock timings (seconds per *one* 6000-step run)

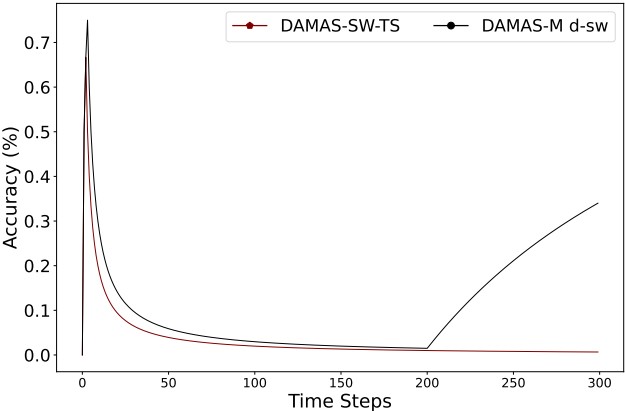

Figure 12: The probability of accurately identifying the current environment over time for DAMAS-Mean d-sw TS and DAMAS-SW TS.

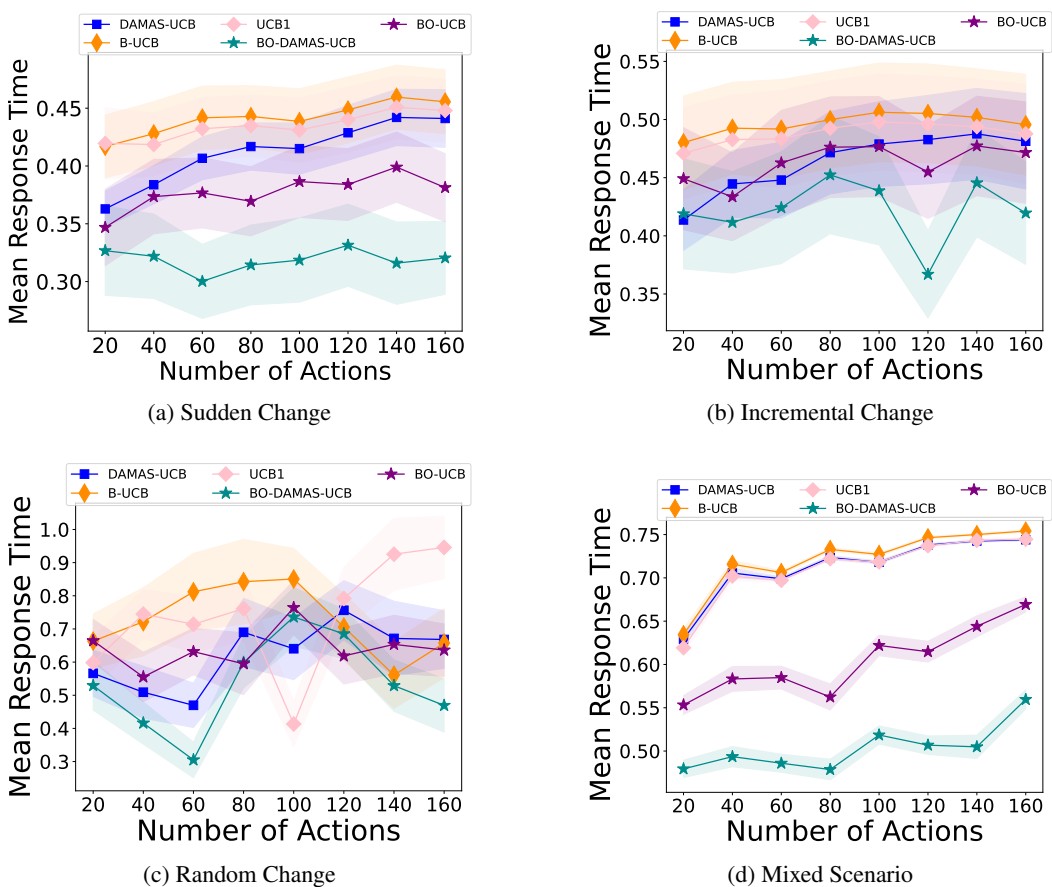

Figure 13: Performance comparison for different environmental change scenarios across varying action counts: Baseline MAB vs. proposed approaches showing average response time.

# B  EXPERIMENT DETAILS

Environmental characteristics and workload patterns (Table 3) are based on the following levels: content type, which represents the format of the data requested; response size, which indicates the

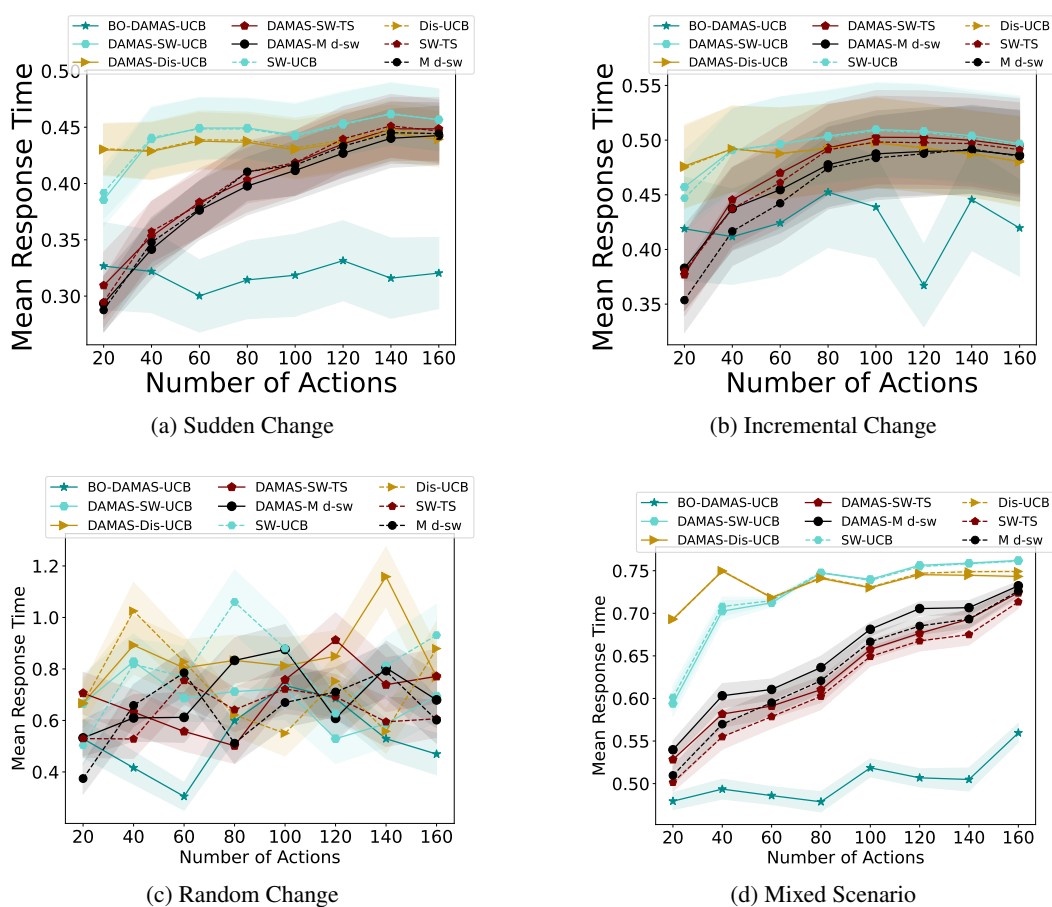

Figure 14: Performance comparison for different environmental change scenarios across varying action counts for multi-agent and single MAB algorithms, showing average response time.

| Actions | DAMAS / UCB1 | Slide / UCB1 | Disc / UCB1 |
|---------|--------------|--------------|-------------|
| 20 | 43 × | 21 × | 0.82 × |
| 40 | 43 × | 38 × | 0.86 × |
| 60 | 41 × | 55 × | 0.88 × |
| 80 | 39 × | 65 × | 0.85 × |
| 100 | 33 × | 69 × | 0.80 × |
| 120 | 41 × | 99 × | 0.82 × |
| 140 | 38 × | 103 × | 0.87 × |
| 160 | 37 × | 114 × | 0.78 × |

Table 2: Overhead factors (vs. UCB1)

amount of data returned in the response; and request entropy, which represents the variability or randomness in the requests. The entropy of the request is categorized as low (repetitive requests) or high (diverse requests). These characteristics are further detailed in Table 4, which provides specific information on the variety and patterns of requests for each content type.

*Real-world web server:* The workload pattern in this experiment represents real-world web server response behaviors under different caching and compression configurations. (e.g., MRU, LRU, LFU, Round-Robin, and File System-based approaches) and compression mechanisms (GZIP with ZLIB and GZ algorithms, both separate and combined with caching methods). as shown in Figures 18, 19 and 20.

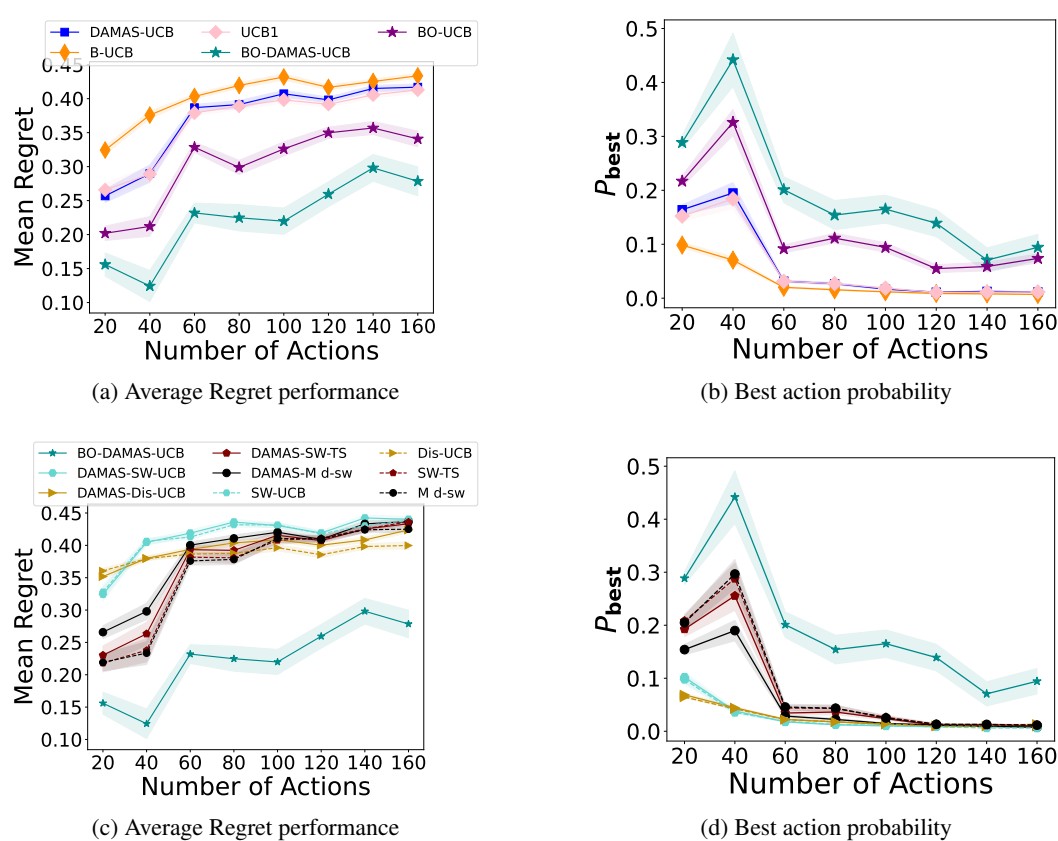

(a) Average Regret performance

(b) Best action probability

(c) Average Regret performance

(d) Best action probability

Figure 15: Performance comparison for Random Change scenario across varying action counts: (a,b) Baseline MAB vs. proposed approaches showing average regret and best action probability; (c,d) Extended comparison across multi-agent and single MAB algorithms. With hazard probability = 1.

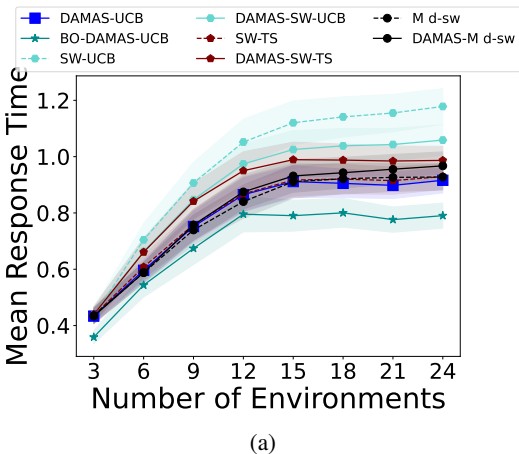

(a)

Figure 16: Average response time of Multi-Agent Approaches and baseline agent under three real workloads.

**Web Server Terminology:**

A **web server** listens for HTTP requests and sends back responses; the *work* (CPU, I/O) changes with what is requested.

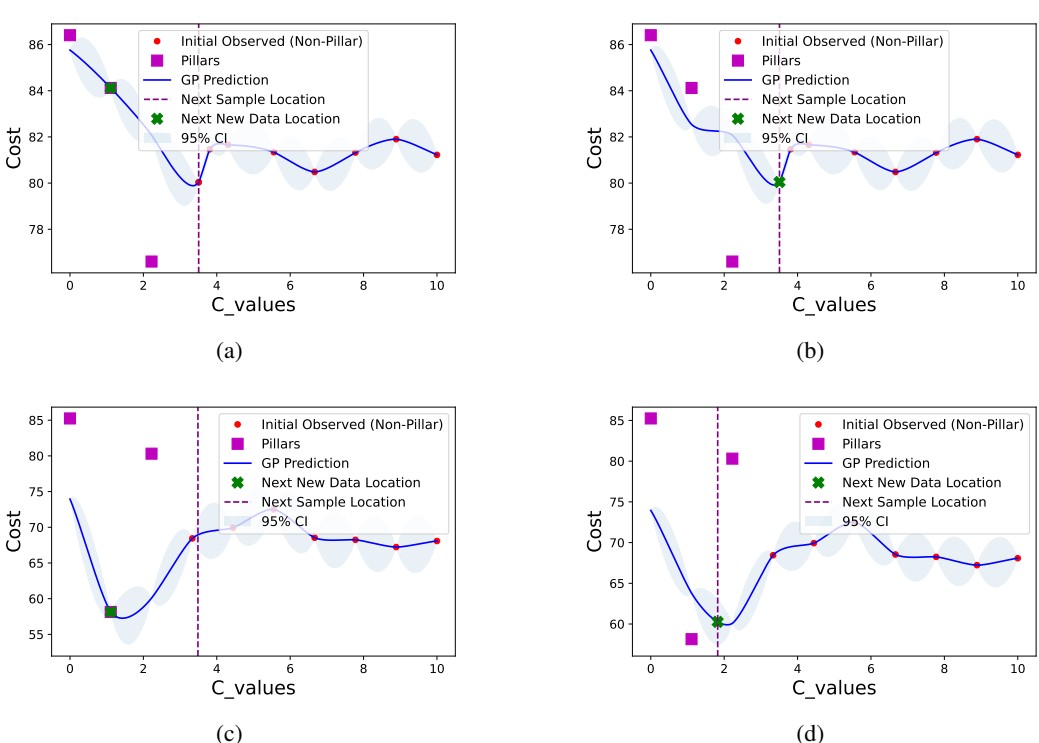

Figure 17: Comparative Analysis of Acquisition Functions: Subfigures (a)-(b) demonstrate the Expected Improvement (EI) acquisition function, indicating the model's preference for regions with anticipated improvements. Subfigures (c)-(d) depict the Lower Confidence Bound (LCB) acquisition function, which explores more diverse input space regions, balancing the exploitation of known areas against the exploration of regions with higher uncertainty. The contrast between EI and LCB illustrates their differing strategies in navigating the search space.

---

**Algorithm 1** DAMAS

---

**Input:** Number of iterations $T$, set of environments $\mathbf{E}$, set of actions $\mathbf{A}$, MAB algorithm $MAB$
**Initialize:**
Create agents $\Phi = \{\phi_1, \phi_2, \ldots, \phi_{|E|}\}$, one for each $e$
Initialize Q-values $Q(\phi, a) = 0$ for all $\phi \in \Phi$, $a \in \mathbf{A}$
Initialize environment probabilities $P(e_i) = \frac{1}{|E|}$ for all $e_i \in \mathbf{E}$
Pre-train each agent on its static $e_i$
**for** $t = 1$ to $T$ **do**
    Sample environment $e_t$ based on current $P(e_i)$
    {Select action $a_t$ using MAB for agent $\phi_t$ corresponding to $e_t$:}
    $a_t = MAB(Q(\phi_t, \cdot), N(\phi_t, \cdot), t)$
    Execute action $a_t$ and observe reward $r_t$
    {Update Q-values for all agents:}
    **for** each $\phi_i \in \Phi$ **do**
        $S(\phi_i, a_t) \leftarrow S(\phi_i, a_t) + P(e_i) \times r_t$
        $N(\phi_i, a_t) \leftarrow N(\phi_i, a_t) + P(e_i)$
        $Q(\phi_i, a_t) \leftarrow S(\phi_i, a_t)/N(\phi_i, a_t)$
    **end for**
    {Update environment probabilities using Bayesian update:}
    **for** each $e_i \in \mathbf{E}$ **do**
        $P(r_t|e_i) = N(r_t; \mu_i(a_t), \sigma_i(a_t)^2)$
        $P(e_i|r_t) \propto P(r_t|e_i) \times P(e_i|r_{t-1})$
    **end for**
    Normalize $P(e_i|r_t)$
**end for**

---

Table 3: Content Types, Sizes, and Request Entropy Characteristics

| Content Type | Size | Request Entropy Example |
|---|---|---|
| HTML | 5 KB (small) | Low: Repeated requests for the same HTML file |
| MP4 (Video) | 300 KB (medium) | Low: Occasional requests for similar video |
| JPG (Image) | 4 MB (high) | High: Varied image requests with randomness |

For a given piece of content the server can choose among **pre-built configurations**, e.g.,

- **Compress-then-send** (smaller packets, extra CPU),

- **Cache-and-reuse** (fast if the same item is requested again).

These choices are *functionally equivalent* (the client still gets the file) but differ in **response-time latency** and resource cost.

Which option is best depends on the **request mix**: compressing JPEG images is wasted effort, while compressing HTML or JSON pays off; caching helps if many clients request the same file, hurts if every request is unique.

Our bandit formulation treats each configuration as an **action** and each distinct traffic pattern (e.g., "image-heavy burst", "static-CSS flood", "API JSON stream") as a **latent environment**.

*"Normalising response times"* in prior work means forcing these latencies—often tens vs. hundreds of ms—into a common 0–1 scale, which can distort decisions when the scales differ by orders of magnitude. *"Known configurations"* simply refers to the finite menu of server tunings listed above; the server can switch among them at will.

DAMAS learns, from raw latency rewards alone, **which configuration works best for the current request**, so that our learning approach to effectively recall the best action when common request sequences recur in future.

Table 4: Workload Characteristics

| Content Type | Variety | Pattern |
|---|---|---|
| HTML | Limited: 11 unique files out of 16 total requests | Consistent: Repeated accesses to certain files |
| MP4 (300 KB) | Limited: 21 unique files out of 40 total requests | Consistent: Noticeable pattern of repeated accesses |
| JPG | High: 40 unique files out of 40 total requests | Random: No discernible pattern in request order |

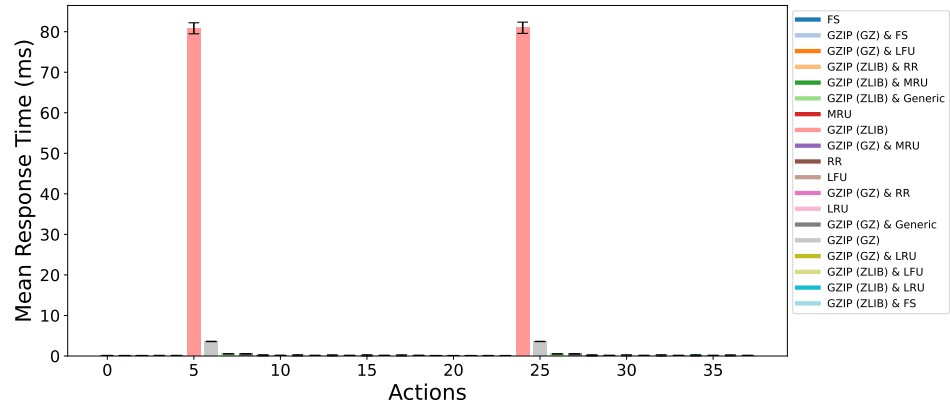

Figure 18: Mean response time for different web server configurations, showing the impact of various caching and compression strategies (HTML).

## C  PROOF OF THEOREM 1

*Proof.* Given the true environment $e^*$, the observed reward $r_t$ is sampled from $P(r_t|e^*)$. This implies that the likelihood (CDF) $P(r_t|e^*)$ is more likely to be larger than that of any other environment $e_j$:

$$P(P(r_t|e^*) > P(r_t|e_j)) > P(P(r_t|e^*) < P(r_t|e_j)), \quad \forall j \neq i.$$

This inequality holds because $r_t \sim P(r_t|e^*)$, making $P(r_t|e^*)$ more likely to explain the observed reward than $P(r_t|e_j)$.

Recursive update of posterior difference: Define the difference between the posterior probabilities of $e^*$ and any other environment $e_j$:

$$\Delta_t = P(e^*|r_t) - P(e_j|r_t), \quad \text{where} \quad P(e_j|r_t) > 0.$$

The posterior probabilities are updated using Bayes' theorem:

$$P(e_i|r_t) = \frac{P(r_t|e_i) \cdot P(e_i)}{\sum_k P(r_t|e_k) \cdot P(e_k)}.$$

Thus, $\Delta_t$ evolves recursively as:

$$\Delta_{t+1} = \Delta_t + \frac{P(r_t|e^*) - P(r_t|e_j)}{\sum_k P(r_t|e_k) \cdot P(e_k)}.$$

Since $P(r_t|e^*) > P(r_t|e_j)$ with high probability, $\Delta_t$ tends to grow over time.

Convergence result: As $t \to \infty$, the cumulative effect of these updates ensures that the posterior probability for $e^*$ converges to 1, while the probabilities for all other environments converge to 0:

$$P(e^*|r_t) \to 1, \quad P(e_j|r_t) \to 0, \quad \forall j \neq i.$$

This confirms that the agent correctly identifies and converges to the true environment in the long run.

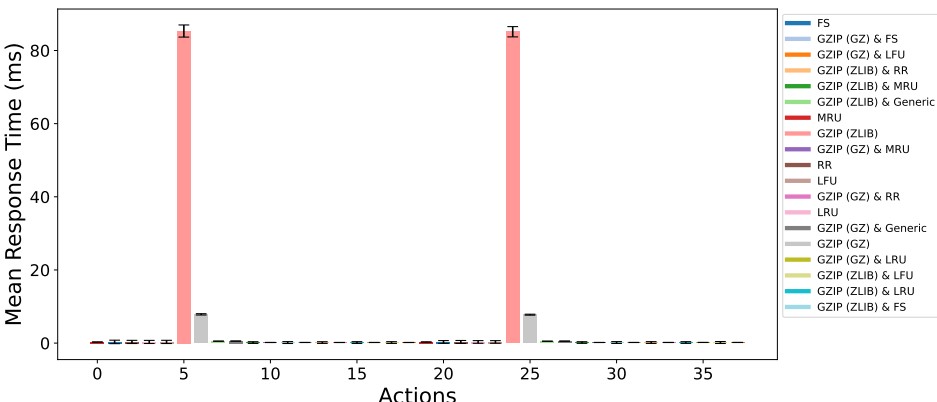

Figure 19: Mean response time for different web server configurations, showing the impact of various caching and compression strategies (MP4).

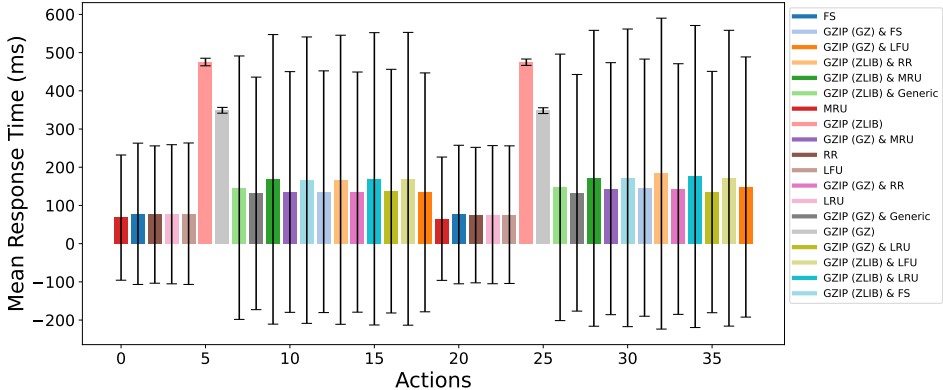

Figure 20: Mean response time for different web server configurations, showing the impact of various caching and compression strategies (JPG).

Concerning reduction to UCB1: Once $P(e^*|r_t) \to 1$, the environment is effectively identified, and the system behaves as if it is operating in a stationary environment $e^*$. The Q-values are updated as:

$$Q(e^*, a_i) = \frac{\sum_{t=1}^{T} r_t \cdot P(e^*|r_t) \cdot I[a_t = a_i]}{\sum_{t=1}^{T} P(e^*|r_t) \cdot I[a_t = a_i]},$$

where $I[.]$ is the indicator function, 1 if the action taken is $a_i$, and 0 otherwise.

Let $T_0$ be an iteration where $P(e^*|r_t) \to 1$ for all $t > T_0$. We can re-write the Q-value update equation as:

$$Q(e^*, a_i) = \frac{\alpha + \sum_{t=T_0}^{T} r_t \cdot I[a_t = a_i]}{\beta + \sum_{t=T_0}^{T} I[a_t = a_i]},$$

where $\alpha = \sum_{t=1}^{T_0} P(e_i|r_t) \cdot r_t$, cumulative weighted reward before convergence, and $\beta = \sum_{t=1}^{T_0} P(e_i|r_t)$, cumulative weighted probability before convergence.

As $T \to \infty$, the influence of $\alpha$ and $\beta$ diminishes, and:

$$Q(e^*, a_i) \to \frac{\sum_{t=T_0}^{T} r_t \cdot I[a_t = a_i]}{\sum_{t=T_0}^{T} I[a_t = a_i]},$$

which is equivalent to the standard UCB1 update.

Concerning regret bound: In the fixed environment $e^*$, the regret of UCB1 is well-known to be:

$$R(T) = O(\log T),$$

therefore our regret is also:

$$R(T) = O(\log T).$$

$\square$

## D    PROOF OF THEOREM 2

*Proof.* **Phase 1: *Within a Single Environment***: The agent starts in environment $e_i$ and uses Bayesian inference to maintain posterior probabilities:

$$P(e_j|r_t) = \frac{P(r_t|e_j) \cdot P(e_j)}{\sum_k P(r_t|e_k) \cdot P(e_k)}.$$

Initially, $P(e_j|r_t)$ is uniform across all $e_j \in E$. At each step $t$, the agent observes $r_t \sim P(r_t|e_i)$. Since the rewards come from $P(r_t|e_i)$, the posterior $P(e_i|r_t)$ increases:

$$P(e_i|r_t) \to 1 \quad \text{and} \quad P(e_j|r_t) \to 0, \; \forall j \neq i.$$

As $P(e_i|r_t) \to 1$, the estimation becomes effectively stationary. The agent updates the Q-values as:

$$Q(a_i) = \frac{\sum_{t=1}^{T_i} r_t \cdot I[a_t = a_i]}{\sum_{t=T_0}^{T_i} I[a_t = a_i]},$$

which is equivalent to the UCB1 update rule in a stationary environment $e_i$. As $T_i \to \omega_i$, the regret is bounded as:

$$R(T) = O(\log T).$$

**Phase 2: Transition Between Environments**: After $T_i$ iterations, the environment transitions from $e_i$ to $e_{i+1}$. The agent experiences a transition period and begins adapting to $e_{i+1}$. Initially, $P(e_{i+1}|r_t)$ may be low. As the agent observes rewards $r_t \sim P(r_t|e_{i+1})$, the posterior probability $P(e_{i+1}|r_t)$ increases, eventually converging:

$$P(e_{i+1}|r_t) \to 1 \quad \text{and} \quad P(e_j|r_t) \to 0, \; \forall j \neq i + 1.$$

**Phase 3: Ordinal Convergence**: Let $\Omega = \{\omega_1, \omega_2, \ldots, \omega_k\}$ be the agent's timeline, where each $\omega_i$ represents a distinct phase corresponding to environment $e_i \in E$. Each $\omega_i$ is a countably infinite phase during which the agent adapts to the current environment $e_i$. Summing the regret across all $k$ environments gives the total regret:

$$R(T) = \sum_{i=1}^{k} O(\log T),$$

$$R(T) = k \cdot O(\log T)$$

By the definition of Big O notation:

$$O(g(n)) = \{f(n) : \exists c, n_0 > 0 \text{ such that } 0 \leq f(n) \leq c \cdot g(n) \; \forall n \geq n_0\}$$

There exist constants $c_i$ and $T_i$ for each environment such that:

$$R_i(T) \leq c_i \cdot \log(T) \text{ for all } T \geq T_i$$

Let $C = \sum_{i=1}^{k} c_i$ and $T_0 = T_k$. Then:

$$R(T) \leq C \cdot \log(T) \text{ for all } T \geq T_0$$

Therefore, $R(T) = O(\log T)$

$\square$

**Theorem 2 convergence:** The empirical results shown in Figures 21 and 22 provide support for Theorem 2's convergence properties in dynamically changing environments. Figure 21 shows that DAMAS exhibits a growth pattern similar to UCB1, which is known to have $O(\log T)$ regret. It is important to note that in our experimental setup, we reset UCB1 at each environmental change. Moreover, Figure 22 illustrates the rapid convergence of environment identification accuracy, which supports the theorem's claim that $P(e_i|r_t) \to 1$ within each environment. This empirical evidence aligns with the theoretical prediction that the agent can effectively identify and adapt to each environment $e_i$ when given sufficient time ($T_i \to \infty = \omega_i$).

Note that this experimental configuration differs fundamentally from our main evaluations, where DAMAS is compared with conventional baselines—such as UCB1—that operate without any access to change-environment information.

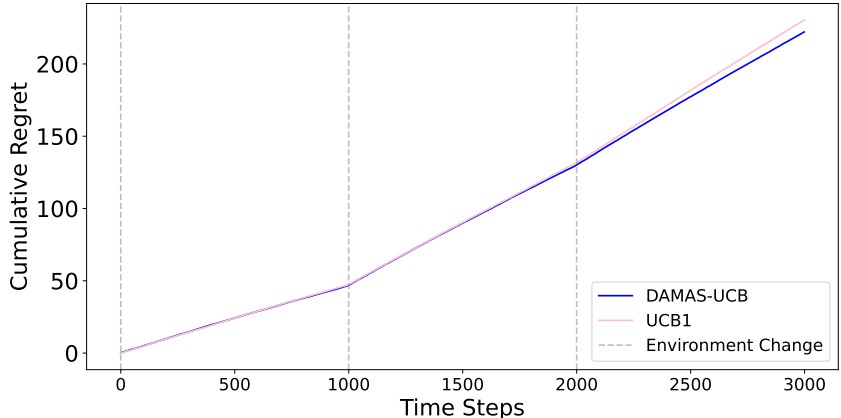

Figure 21: Comparison of DAMAS and UCB1 in Dynamically Changing Environments

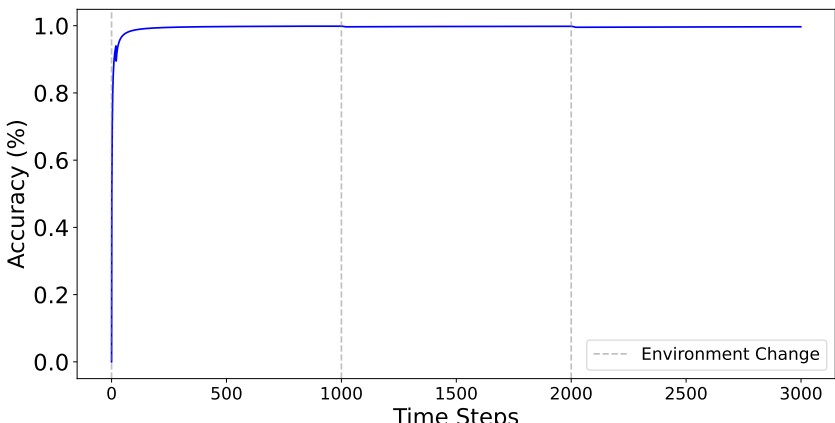

Figure 22: The probability of accurately identifying the current environment over time

# E    ADDITIONAL RELATED WORK

Several algorithms tackles non-stationary bandits via changepoint detection and restarts, such as GLR-klUCB (Besson et al., 2022) and M-UCB (Cao et al., 2019), which couple UCB indices with GLR or window-based detectors to reset statistics after detected changes. Rexp3 (Besbes et al., 2019) addresses non-stationary rewards by running an Exp3 algorithm over batches and resetting or reinitializing weights across batches. In all these approaches, adaptation is achieved by forgetting past experience. By contrast, DAMAS maintains a pool of specialized agents and updates their value

estimates continuously; changes are handled via Bayesian reweighting of environment probabilities rather than explicit resets. When an environment reoccurs, DAMAS can immediately exploit the corresponding agent instead of re-learning from scratch.

Restless bandits extend classical MABs by modeling each arm as a Markov process that evolves continuously, usually as an MDP or a Markov chain for each arm (Niño-Mora, 2023; Wang et al., 2020). However, our non-stationary bandit model is fundamentally different. That is, DAMAS does not assume Markovian state dynamics at the arm level, and hence we do not consider that agents can observe arm states and then infer transition functions. Additionally, we do not consider that agents actions interfere with the environment transitions. In our model, environments change due to external and unknown reasons, and our focus is on estimating the current environment in a context-free manner.

## F    CHALLENGES ON CONTEXTUAL DEPENDENT METHODS

In this work we focus on a reward driven approach, and one might naturally ask how this compares to methods that also exploit contextual information. Context can indeed be highly informative in many applications and may improve decision quality. However, in other domains, contextual information can be noisy or difficult to obtain. In this section, we investigate the impact of noisy context on contextual bandit methods, highlighting the challenges that arise when the available features are noisy.

Li et al. (2010) demonstrated the effectiveness of a contextual multi-armed bandit solution for personalized news recommendation. Building on this framework, we introduce controlled noise into the contextual features to examine its impact on the learning policy. As shown in Figure 23, increasing noise levels degrade overall performance, highlighting how noisy or unreliable information can adversely affect decision-making.

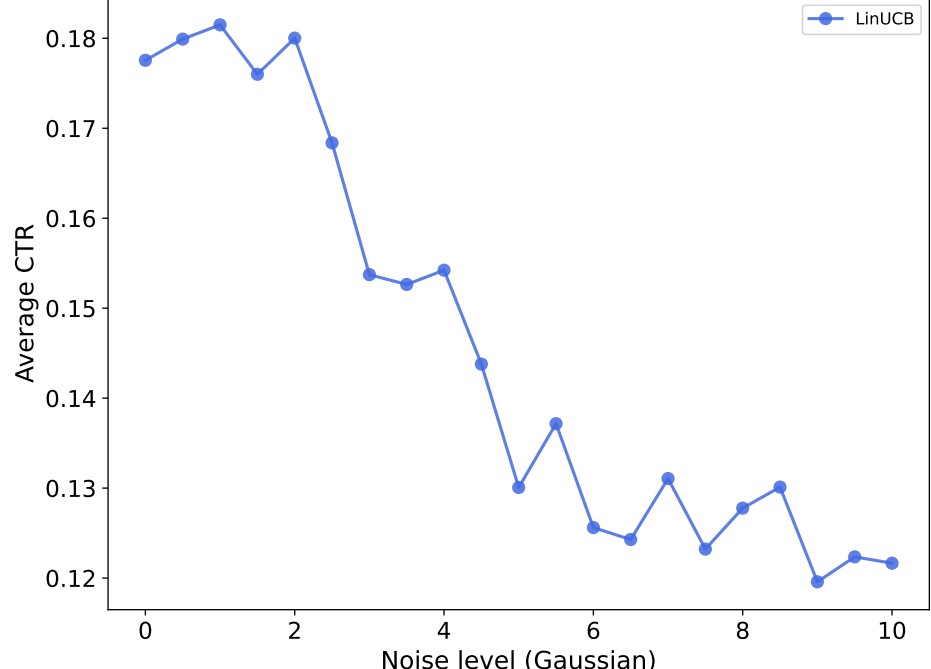

Figure 23: LinUCB contextual bandit under gradually increasing noise in the feature vectors, showing degraded performance at higher noise levels.

# G    ADDITIONAL EXPERIMENTS: REWARDS OVERLAPPING ACROSS ENVIRONMENTS

In the main paper, we considered several scenarios in which both the underlying environment and the reward ranges change over time. However, in many practical settings, the environment may evolve, while the reward ranges across environments remain similar or even highly overlapping. In this section, we therefore conduct additional experiments that focus on such cases, examining how our approach and the baselines perform when environments change but their reward ranges are largely aligned.

These environments are generated at random to provide a more comprehensive assessment. As illustrated in Figure 24, the random generation process produces dynamic environments with the same reward range.

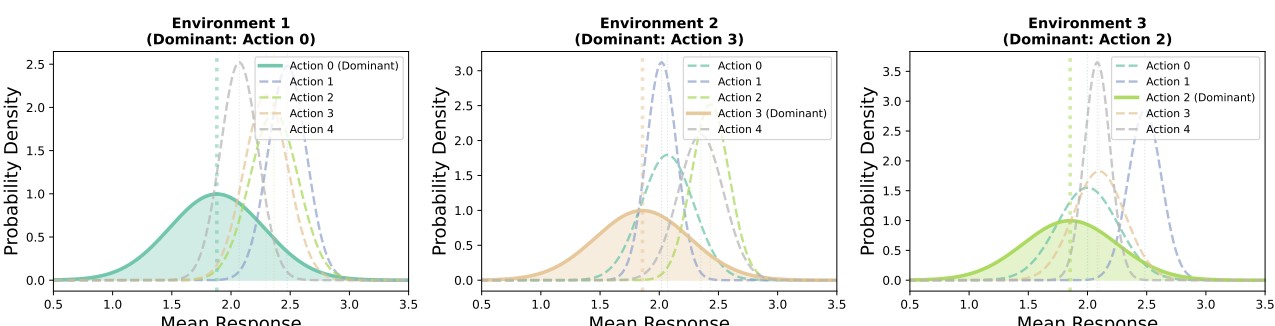

Figure 24: Randomly generated overlapping reward distributions in a same-range workload

BO-DAMAS-UCB as shown in Figure 25, outperforms other methods on both metrics (mean regret and best action selection), even as the number of actions grows.

As depicted in Figure 26 with highly overlapping reward ranges across environments, BO-DAMAS-UBC consistently maintains lower regret and higher best-action probability than competing methods in all environment counts, with a fixed number of agents (4 agents).

As shown in Figure 27c and 27d, BO-DAMAS-UCB outperforms all single-agent baselines in this setting with 4 environments and 3 agents. Across all numbers of actions, BO-DAMAS-UCB attains lower mean regret and high probability of action selection than GLR-klUCB, M-UCB, and RExp3.

Moreover, in Figure 28, with 3 environments and only 2 agents (where the third environment is an interpolation of the first two), BO-DAMAS-UCB again achieves the lowest mean regret across all numbers of actions and maintains a higher probability of selecting the optimal action than all baselines.

## LLM USAGE:

Editing (e.g., grammar, spelling, word choice, paraphrase), Understanding technical concepts, and Producing some LaTeX format for the tables.

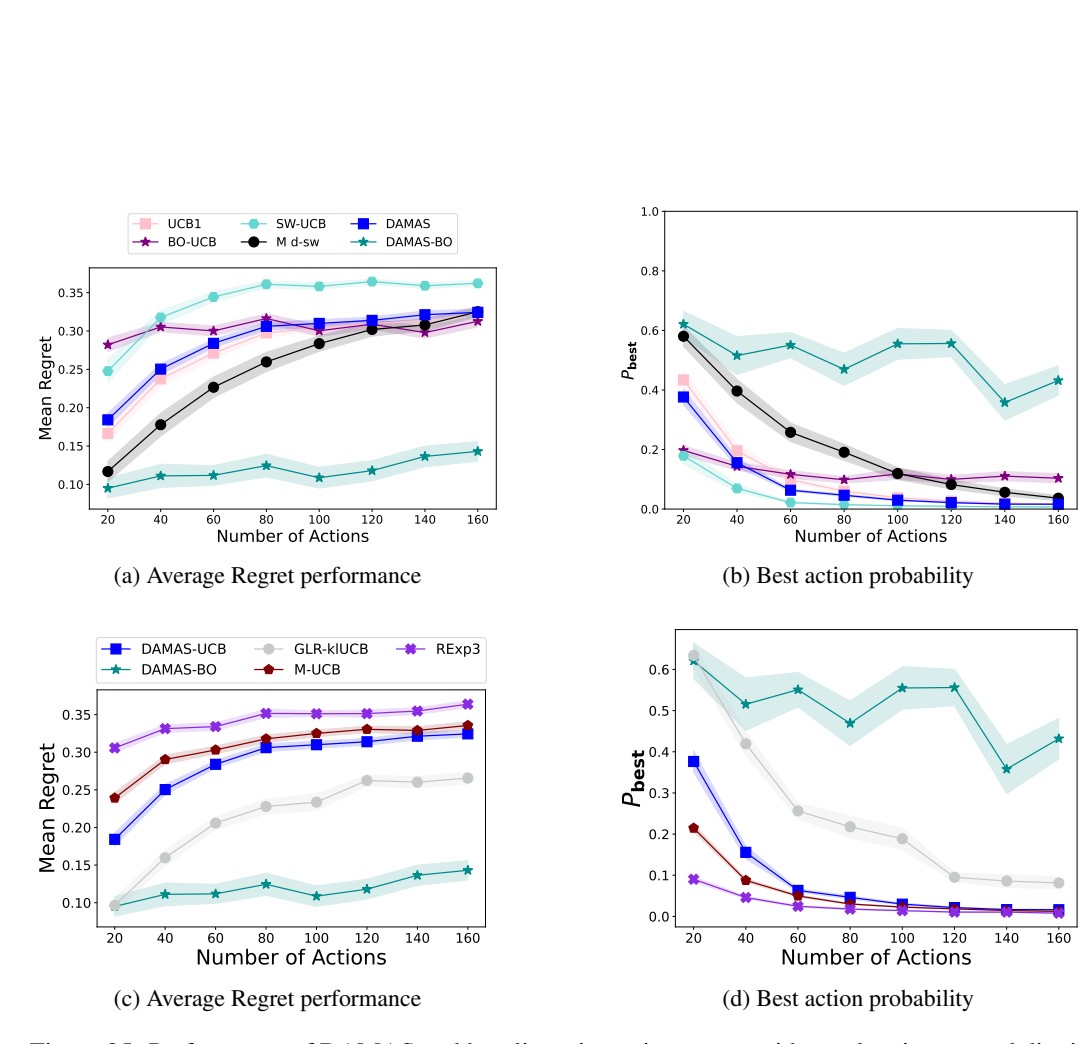

Figure 25: Performance of DAMAS and baselines, in environments with overlapping reward distributions. (a, c) Average regret across actions; (b, d) Probability of selecting the optimal action.

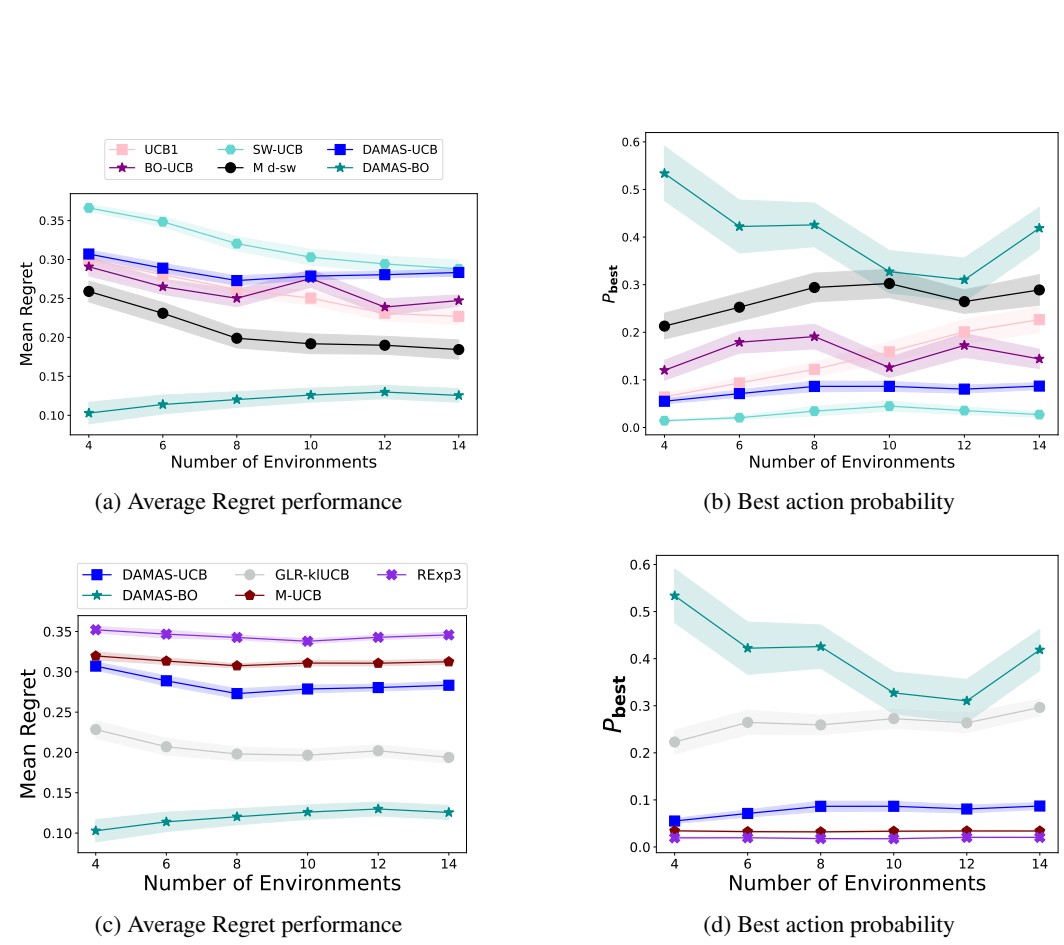

Figure 26: Performance of DAMAS and baselines in environments with overlapping reward distributions, as the number of environments increases while the number of agents is fixed to four. (a) and (c) show average regret across 80 actions; (b) and (d) show the probability of selecting the optimal action.

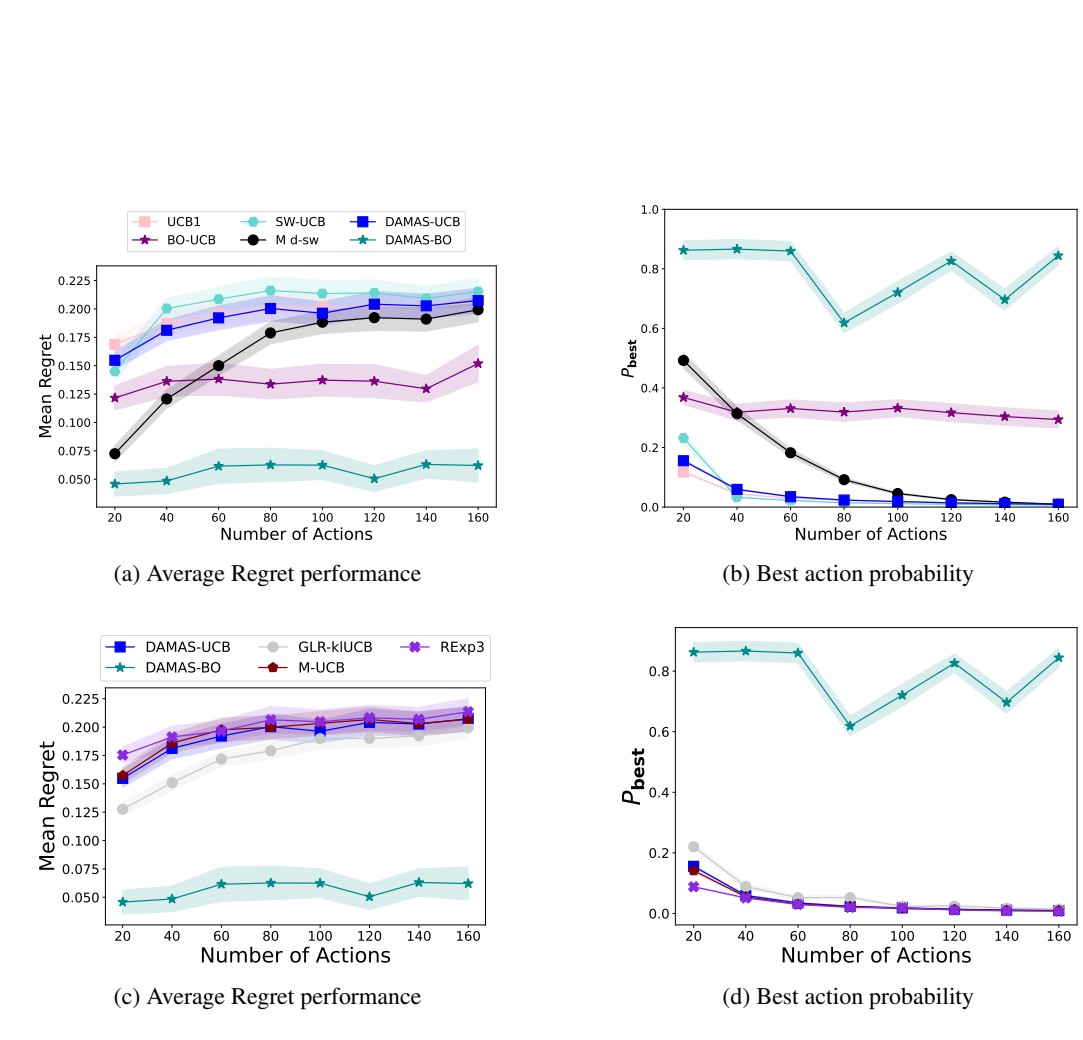

Figure 27: Performance of DAMAS and baselines, with 4 environments and 3 agents. (a, c) Average regret across actions; (b, d) Probability of selecting the optimal action.

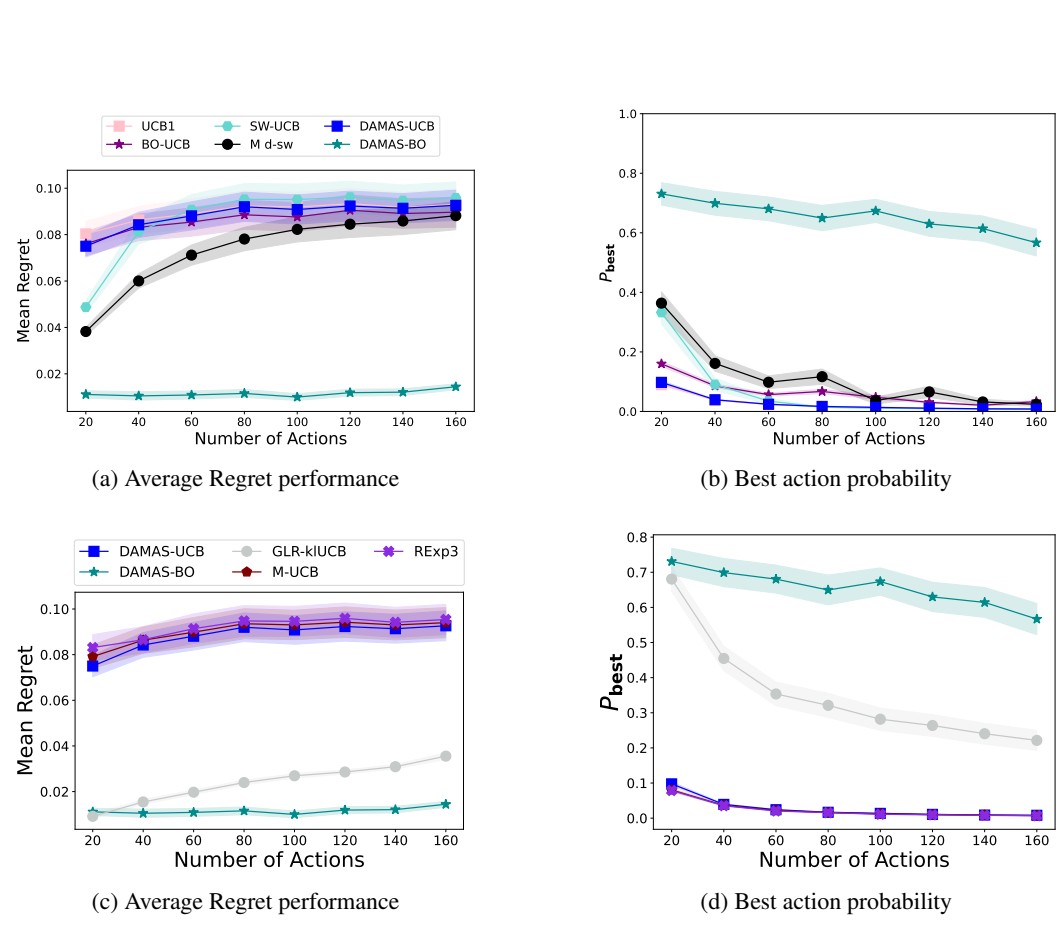

Figure 28: Performance of DAMAS and baselines, with 3 environments and 2 agents, and the third environment is an interpolation between the first 2 environments. (a, c) Average regret across actions; (b, d) Probability of selecting the optimal action.

