# OpenReview forum: "A Bayesian Multi-agent Multi-arm Bandit Framework for Optimal Decision Making in Dynamically Changing Environments"
_ICLR.cc/2026/Conference — Submitted to ICLR 2026_

### Official Review · Reviewer_L1tZ · 2025-10-26

**Soundness:** 2
**Presentation:** 2
**Contribution:** 2
**Rating:** 2
**Confidence:** 2

**Summary:**

This paper proposes DAMAS, a Bayesian multi-agent multi-armed bandit framework for non-stationary environments. Each agent specializes in a latent environment, and the system maintains Bayesian beliefs over which environment is active based on reward observations without using explicit contextual features. DAMAS integrates Bayesian updates for environment inference and Bayesian optimization for adaptive exploration tuning. The authors provided a theoretical regret analysis, and conducted numerical experiments on both synthetic and real-world data to show the effectiveness of their method.

**Strengths:**

- The paper proposes a framework that integrates multi-agent modeling, Bayesian environment inference, and MAB decision-making, which is conceptually novel. I also think this kind of problem has practical relevance as it addresses dynamic systems without contextual signals.
- The paper provides a detailed empirical evaluation across multiple types of non-stationarity.

**Weaknesses:**

- One major issue with this paper is in its evaluation metrics, as there appears to be a mismatch between theoretical and empirical regret definitions here. If I understand correctly, the theoretical analysis establishes regret bounds relative to a static optimal arm within each stationary phase (as in UCB), whereas the experiments compute regret against a dynamic oracle that knows the best arm for each environment at every time step. This inconsistency makes it difficult to interpret how the theoretical guarantees relate to the reported empirical improvements, and limits the clarity of what kind of optimality DAMAS actually achieves. The authors were also not very clear what is the benchmark they are comparing against.
- Related to the point above, the main theoretical results (Theorems 4.1 & 4.2) essentially reduce to showing that DAMAS converges to UCB1-style behavior within stationary phases, yielding O(logT) static regret. This result relies on long dwell times and offers little advance over classical stationary-bandit analyses. I think for a non-stationary bandit setting, it'd be much more helpful to investigate the dynamic regret, to enable comparison with related works on non-stationary bandits.
- I also still have concerns over the computational overhead of the current framework. The multi-agent Bayesian updates and Bayesian-optimization tuning is reported as roughly 39.3× slower than UCB. For real-world, large-scale or high-frequency decision systems, this can be too much to handle and further optimization might be required.
- The paper mainly considers UCB/TS-based methods as its baselines, but it omits standard variation-budget and parameter-free non-stationary bandit methods such as RExp3.

**Questions:**

- The framework requires pre-training each agent in a fixed environment to estimate, then uses Gaussian likelihoods to update environment posteriors online. I wonder if this framework can remain robust if the real non-stationarity deviates from those pre-trained distributions, or when the number/identity of environments is misspecified?

---

> ### Author Response · Authors · 2025-12-04
>
> - **Mismatch between theoretical and empirical regret definitions:** We thank the reviewer for this point, and we would like to say that there was a potential confusion about this.
>
>   In this paper, both the theoretical analysis and the empirical evaluation use a dynamic regret notion: the comparator is always the best arm for the current environment, which can change over time (as environments change). That is, at each time step, the environment may be different from previous steps, and therefore the reward distribution of each arm may change. The optimal arm is defined locally as the arm with the highest expected reward under the environment that is active at that time. Dynamic regret then measures how much cumulative reward is lost compared to an oracle that, at every time step, selects this best arm for the current environment.
>
>   Under this definition, **there is no conceptual mismatch between the theoretical and empirical parts:**
>
>   - In the theoretical analysis, we consider a piecewise stationary setting where the environment is fixed within a phase, then may change to a different environment in the next phase. Within any given phase, the best arm for that environment is constant. Theorems 4.1 and 4.2 show that, once the Bayesian inference in DAMAS has identified the current environment, the algorithm behaves like a UCB1 learner and achieves logarithmic regret in that phase.
>   - In the empirical evaluation, we use exactly the same comparator: at each time step we compare the algorithms to an oracle that knows which environment is currently active, and compute the regret based on the played arm and the mean value of the current best arm. The reported regret curves are therefore measuring dynamic regret across the entire non‑stationary time, including frequent and abrupt changes.
>
>
> - **UCB1 within phases and dynamic regret in non stationary setting:** Perhaps there was a confusion, since we are evaluating the dynamic regret. In Theorems 4.1 and 4.2 we show that, once the current environment has been identified, DAMAS converges to UCB1 style behavior within that stationary phase and achieves logarithmic regret with respect to the best arm for that environment during that phase. Interpreted under the dynamic regret notion above, these are dynamic regret bounds within each phase: within a phase, the environment and thus the optimal arm do not change, but the optimal arm may (and often will) change in the next phase, when the environment changes.
>
>   We also provide additional support to the idea that DAMAS converges to the UCB1 policy. As shown in Figure 21 (Appendix D), in non-stationary settings we restart UCB1 in each phase, while DAMAS dynamically identifies the current best action. Over time, its behavior converges toward that of the UCB1 algorithm.
>
>
> - **Expanded Baselines:** We extended our experiment and compare our approach to other non-stationary bandits algorithms including: **GLR-klUCB, M-UCB and Rexp3.** Our approach outperforms these methods **(Appendix G Figures25, 26, 27 and 28).** We also included a discussion on how these algorithms differ from our approach **(See Appendix E ADDITIONAL RELATED WORK).**
>
> - **Pre-trained agents and mismatch number of environments/agents:** Appendix G presents further results where the number of deployed agents do not match the true environments and the runtime environments are unseen during pre-training. DAMAS adapts by weighting the posterior across candidate agents, allowing those closest to the runtime conditions to specialize over time. In all cases, DAMAS outperforms single-agent and other adaptive baselines.

---

### Official Review · Reviewer_kXPd · 2025-10-30

**Soundness:** 3
**Presentation:** 3
**Contribution:** 1
**Rating:** 2
**Confidence:** 2

**Summary:**

The paper investigates a multi-state multi-armed bandit setting by proposing an algorithm that dynamically estimates the probability of being in each state. Each state is characterized by changing reward distributions and dynamic constraints. Theoretical guarantees are provided with respect to finite time regret performance of the algorithm. Empirical results also demonstrate some improvement over proposed baseline algorithms.

**Strengths:**

- This paper addresses relevant problems in non-stationary MAB with valid application areas, such as server allocation, communications etc.

- This paper provides a good amount of empirical evidence to support their claims in comparison to the benchmarks the authors designed.

**Weaknesses:**

- Seems to me like the framework considers a MAB problem where the states, which govern the reward space is evolving over time. I have a hard time discerning the difference between this model and that of a non-stationary multi-armed bandit model captured by the restless bandit framework (espcially refering to lines 130-137). Even if there are (perhaps marginal) differences between the authors problem setting and the restless bandit setting, it seems like this should have been at least mentioned in the literature review.

- I have a hard time understanding the multi-agent aspect of this work. It seems to me that the algorithm is finding the best agent for the context, and each agent accordingly takes the best action. How is this any different than regular MAB with nested action space? That is we consider the "agent" in this paper just another "action" - could we not also solve this using combinatorial bandit for example? If so then the authors should consider a larger array of baseline algorithms.

- W.r.t. to the theory, the authors argue that their problem reduces to a UCB algorithm (if I'm not mistaken), and given that UCB has logarithmic regret, so does their algorithm. However, there is more to be desired here, and I encourage the authors to consider per se, whether the algorithm suffers from gap-dependent or gap-free regret, scaling w.r.t. the number of agents, and size of the action space etc. Overall w.r.t. to bandit theory, it is lacking some richness and depth here. It also ties back to my previous point regarding the multi-agent aspect, if this is just UCB with more arms and changing contexts - previous work has shown that in these cases, regret is logarathmic in T [1, 2].

[1] Tekin, Cem, and Mingyan Liu. "Online learning of rested and restless bandits." IEEE Transactions on Information Theory 58.8 (2012): 5588-5611.

[2] Liu, Haoyang, Keqin Liu, and Qing Zhao. "Logarithmic weak regret of non-bayesian restless multi-armed bandit." 2011 IEEE International Conference on Acoustics, Speech and Signal Processing (ICASSP). IEEE, 2011.

**Questions:**

- Are there any interactions between agents and their actions, when acting cooperatively? i.e. are the rewards w.r.t. actions submodular/supermodular?

- Do the agents have any sense of "agency", or are they purely controlled by the central controller? In which agents may be free to act whatever way they choose under the circumstance and reward structure given - adding complexity to the decision process.

---

> ### Author Response · Authors · 2025-12-04
>
> - **Restless bandits:**
>   While both frameworks address non‑stationarity, there are **fundamental differences**. Restless bandits assume arms evolve according to a stochastic process (typically Markovian). In contrast, **DAMAS models non‑stationarity at the environmental level**, using specialized agents and Bayesian inference rather than explicit Markovian per‑arm state processes. Hence, we do not observe the state of the arms, and do not try to infer an arm-based transition function. Additionally, we do not consider that the actions of our agent would affect the environment transitions (as assumed in some of the restless bandits frameworks). **We added a discussion of restless bandits to the ADDITIONAL RELATED WORK section (Appendix E) to clarify these distinctions.**
>
>
>
> - **Regular MAB (nested action) and combinatorial bandits:**
>   MABs with nested action spaces and combinatorial bandits aim to learn **structured actions**, for example in a hierarchical way. DAMAS, however, adapts to **changing environments** by aggregating over agent specializations with Bayesian inference. While nested action MABs and combinatorial bandits structure the **choices**, DAMAS structures the **learning entities (agents)**, focusing on environment switching rather than action structure.
>
>
>
> - **Multi‑agent interactions:**
>   DAMAS follows a **mixture‑of‑experts style architecture** rather than a fully distributed multi‑agent system. The framework maintains multiple specialist agents, each tailored to a particular environment, with its own Q‑function and policy that outputs an action recommendation. A centralized Bayesian controller acts as the gating mechanism: it maintains posterior beliefs over which environment is currently active, combines the agents’ recommendations, and probabilistically selects one agent’s proposed action according to these beliefs. After executing the chosen action and observing the reward, the controller updates every agent’s Q‑values, weighting each update by its posterior probability of being the active environment.
>   In short, agents are **experts whose advice is sought by the central system**, rather than fully distributed peers. Each agent maintains separate value estimates, but only one controls the actual decision at each step.

---

### Official Review · Reviewer_RXpQ · 2025-10-31

**Soundness:** 3
**Presentation:** 4
**Contribution:** 2
**Rating:** 6
**Confidence:** 3

**Summary:**

This paper proposes Dynamic Adaptation through Multi-Agent Systems (DAMAS), a new method for non-stationary multi-armed bandits (MABs). DAMAS does not require any contextual information, only observing rewards. The problem is modeled as a set of $E = \{e_0, \dots, e_n \}$ MABs (where the rewards for each arm are sampled from a Gaussian distribution with unknown mean and variance). It is unknown from which of the $e_i, i \in [0,n]$ MABs the sampled reward comes from. The non-stationarity is defined as the change from one $e_i$ to another $e_j$. DAMAS instantiates one agent $\phi_i$ for each environment $e_i$. Additionally, it estimates the likelihood $P(r_t|e_i)$ of an observed reward $r_t$ coming from a specific environment $e_i$. Based on this, it selects the corresponding agent, and pulls an arm using UCB. All agents are updated at each timestep, where the update takes into account $P(e_i)$.

**Strengths:**

- The problem is well defined, and based on practical applications (web server workloads)
- The proposed approach is justified, and its efficiency demonstrated on multiple non-stationarity scenarios
- I found the weighted Q-value update (line 161)

**Weaknesses:**

- The framework assumes that 1) the number of environments $|E|$ is known in advance, so it can instantiate the correct number of agents, and 2) that the properties of each $e_i$ are known, since the agents are first trained on the individual environments separately. In a sense, the optimal arm for each environment is known in advance. These seem to me very strong assumptions, that contradict the claim that DAMAS does not rely on external context.
- The limiting assumptions are conveyed in the real-world experiments (Appendix A), where DAMAS has a higher regret and response time than the single-agent counterpart.
- Even though the theoretical contributions are sound, they seem to me of limited use, as they assume that the agent remains in the current environment for long enough ($T_i \rightarrow \infty$).
- Since only the reward is used to estimate the probability of being in a specific environment, the range of rewards need to be different. This is shown in line 301, where arm-means range in $[0.2,0.7], [0.02,0.09], [1.5,2.5]$ for $e_O, e_1, e_2$, respectively. I believe it would be hard for DAMAS to systematically pull the optimal arm when the ranges of rewards are highly overlapping across environments.

Although the algorithm is sound, well justified and clearly explained, to my understanding, it is based on strong assumptions, limiting its use in practice.

**Questions:**

Although to my understanding, the assumptions seem strong. Could you clarify how realistic they are?

---

> ### Author Response · Authors · 2025-12-04
>
> - **Pre-trained agents and generalization:**
>   While DAMAS uses a finite set of pre-trained agents as priors over some environments, it does **not assume any knowledge of the true environments or their optimal arms**. At deployment, the environment may differ from all pre-trained environments. The underlying environments at test time were **randomly generated at runtime**. We also present new results **(Appendix G Figures 26, 27 and 28)** where (i) the number of deployed agents does not match the true number of environments, and (ii) the runtime environments are not present in the pre-training set. In these cases, DAMAS **dynamically selects or interpolates** by weighing the posterior over all candidate agents. The agents closest to the runtime environment gradually improve their Q-values and become more specialized. Across these settings, DAMAS **consistently outperforms baseline methods**, including single-agent and other adaptive models.
>
>
>
> - **Real-world experiments:**
>   In certain real-world settings, the multi-agent extension (DAMAS) to other MAB algorithms exhibits slightly higher regret and response time than its single-agent counterpart. This behaviour arises when the environment identification component is imperfect, occasionally assigning traffic to a suboptimal specialized agent. However, across both synthetic data and real-world web-server workloads, the proposed **DAMAS-BO framework consistently outperforms standard single-agent bandit baselines**.
>
>
>
> - **Theoretical assumption:**
>   Our theoretical results are derived under the assumption that the environment remains in a given state for a sufficiently long period. This assumption is common in non-stationary bandit analysis, as it allows one to meaningfully define regret with respect to an environment-specific optimal arm and to show convergence before a change occurs. We just formalise it in a slightly different way by using the notion of ordinal numbers. Importantly, these assumptions are intended to provide **formal guarantees in an analytically tractable setting**, not to restrict applicability. In practice, DAMAS does **not rely on this assumption**. Its Bayesian update mechanism estimates environment probabilities dynamically from observed rewards at every step, enabling adaptation to **unexpected or frequent changes**. Our experiments confirm this: even when the environment changes randomly at every time step (**hazard probability 1.0, Appendix A Figure15**), DAMAS achieves **lower regret than UCB1, SW-UCB, and other baselines**.
>
>
>
> - **Reward distribution overlap:**
>   We acknowledge the reviewer’s point that highly overlapping reward distributions make identifying the optimal arm more challenging. **However, as shown in our new experiments (Appendix G Figures25 and 26)**, DAMAS maintains **strong performance in both mean regret and probability of selecting the best action** under this suggested setting, even as the number of actions increases as well as the number of environments increases.

---

### Official Review · Reviewer_HB9R · 2025-10-31

**Soundness:** 2
**Presentation:** 3
**Contribution:** 2
**Rating:** 4
**Confidence:** 4

**Summary:**

The authors consider a multi-armed bandit problem with unbounded rewards and non-stationary environments (i.e., there being $ n $ different reward distributions and the distributions changing over time arbitrarily). The authors propose an algorithm called DAMAS that instantiates $ n $ different instances of a multi-armed bandit algorithm - one for each environment - and uses Bayesian posteriors for updating the algorithms' internal states (e.g., value estimates, etc.) and the posterior probabilities of being in each environment. The authors show a phase-wise asymptotic logarithmic regret bound (i.e., under the assumption of spending an infinite amount of time in each environment before transitioning) and strong experimental results on synthetic setups and real-world web server workloads.

**Strengths:**

* The paper studies an important sub-area of multi-armed bandits: non-stationary environments and unbounded rewards.

* The paper proposes a simple and principled way of dealing with non-stationary environments, namely, maintaining and updating Bayesian posteriors over the environments.

* The experiments on synthetic setups cover a range of non-stationary characteristics (environments changing abruptly, gradually, randomly, etc.) and show strong empirical performance of DAMAS (the proposed algorithm) over baselines.

**Weaknesses:**

*  The regret bound is quite weak because it requires the algorithm to spend an infinite amount of time in each environment before transitioning. Not only is the regret bound asymptotic (i.e., it does not hold for a fixed time horizon), it requires each phase length to be asymptotic as well.

* The paper motivates its use of reward and no other context as a strength because it requires less information. However, I think this is a weakness because context is present and is important in many applications. Many other lines of work in the multi-armed bandit literature started out with the non-contextual setting and over time included the contextual setting because it is prevalent in many applications (e.g., bandits with knapsacks -> contextual bandits with knapsacks in online advertising and resource allocation problems). It would be fair to say that considering only rewards is a choice to simplify the problem. However, I don't think it's fair to say that this work overcomes the difficulty of obtaining contextual features.

**Questions:**

Please see my comments in the weaknesses section.

---

> ### Author Response · Authors · 2025-12-04
>
> - **Regarding the asymptotic regret bound:**
>   We would like to emphasize three key points:
>   1. **Common assumption in the literature:** Asymptotic regret bounds are very common in the MAB literature. Additionally, assuming piece-wise stationarity is also common in the non‑stationary bandit literature when providing environment‑wise regret guarantees [1]. However, we formalise the notion of piece-wise stationarity in a slightly different way by using ordinal numbers.
>
>    2. **Not required by the algorithm:** DAMAS can run without any knowledge of phase lengths or change times. The Bayesian updates operate at every step regardless of how long each environment persists.
>
>   3. **Empirical focus on finite phases:** Our evaluation explicitly targets the more challenging and practically relevant case where phase lengths are finite and can be very short. In particular, we include experiments with frequent and abrupt changes, including hazard probability equal to 1.0 (i.e., the environment can change at every time step), **as shown in Appendix A Figure15**.
>
>
>   [1] Liu, Fang, Joohyun Lee, and Ness Shroff. *“A change‑detection based framework for piecewise‑stationary multi‑armed bandit problem.”* AAAI Conference on Artificial Intelligence, Vol.32, No.1, 2018.
>
>
>
> - **Contextual information:**
>   We agree that contextual information is important in some domains and can be incorporated using several methods. However, there are other domains where **high‑quality context is unavailable, unreliable, or costly to obtain**. For instance, this challenge has been explicitly modeled in recent work on bandits and contextual bandits with costly or noisy features [2].
>
>   To illustrate this empirically, we include an experiment where a standard LinUCB contextual bandit is run on the same dataset while we gradually add noise into the feature vectors. **As the noise level increases, the performance of LinUCB degrades substantially**, highlighting that low‑quality or expensive context can negatively affect overall performance (**Appendix F Figure 23**).
>
>   Hence, there is value in a context-free method, and it may be more suitable than context-based approaches in some domains. We will adjust our discussion accordingly.
>
>   [2] Kim, Jung‑Hun, et al. *“Contextual linear bandits under noisy features: Towards Bayesian oracles.”* International Conference on Artificial Intelligence and Statistics. PMLR, 2023.

---

### Meta-Review · Area_Chair_ytwZ · 2025-12-16

**Summary:**

This paper investigates an important setting of non-stationary bandits with changing reward distributions, and proposes a Bayesian "mixture-of-experts" style multi-agent framework (DAMAS) that maintains posterior beliefs over latent environments and routes decisions to specialized agents. The performance of the proposed DAMAS framework is further validated via experiments showing empirical gains on synthetic settings and a web-server workload.

Despite the important of the settings, reviewers raised several key limitations and concerns regarding the proposed framework such as
- The theoretical analysis in Section 4 is superficial and somehow misaligned with the claimed setting as studied in this work. In particular, one reviewer argues that the regret guarantee is "quite weak", which requires asymptotic phase lengths and effectively infinite time per environment. Thus this does not hold for realistic finite horizons. Another reviewer questions whether the main theorems reduce essentially to "UCB1-style behavior within stationary phases," as claimed by the authors. This in turn flags a mismatch between theoretical regret definition and empirical evaluation.
- There are also key concerns on the assumptions and the practicality of the proposed framework. Specifically, reviewers question whether the proposed framework DAMAS presumes knowing the number of environments, whether environments oroptimal arms are effectively known via pretraining, and whether DAMAS remains robust when reward distributions overlap or the runtime environment deviates from pretraining. These are key to the applications of the proposed framework and remain unclear.
- There are also concerns on the positioning of this work compared to existing works on non-stationary RL/bandits. In particular, one reviewer finds the difference from existing frameworks such as restless bandits and other non-stationary models to be very unclear. Reviewer suggest the authors to have a clear picture on the differences of the settings studied in this work, and the key contributions and novelty.

**Reviewer Concerns:**

One reviewer raised the questions that the theory uses environment-wise regret while experiments compare to a dynamic oracle. In the rebuttal, the authors clarify that both are intended to use a dynamic regret notion, i.e., the best arm for the current environment at each time. The authors further explain that in the analysis, they consider piecewise-stationary segments and achieve UCB-like behavior within segments. They state that they added new baselines and discussions to align comparisons.

In addition, one reviewer questioned that whether the "multi-agent" aspect is just to select among experts. In the rebuttals, the authors explicitly frame DAMAS as a mixture-of-experts architecture with a central Bayesian controller gating among specialized agents. This clarifies that agents are not fully distributed peers.

Despite the rebuttal, several key concerns or limitations remain in this work, including but not limited to
- Although the authors clarify the intended regret notion and add finite-phase experiments in the rebuttals, the core critique that the formal bounds are asymptotic, phase-based and may not meaningfully characterize frequent-switching non-stationarity remains a potential blocker for theory-focused reviewers. This also limit the technical concerns or contributions of this work.
- This paper’s conceptual framing of multi-agent is criticized as essentially a gated ensemble of bandit learners. The relationship to established non-stationary models such as restless bandits, variation-budget, or parameter-free, may remain unconvincing even after additional related-work discussion. In the revision, it is suggested to the authors to have a clear picture about the position of this work compared to many existing works in this domain. This will be helpful to highlight the novelty and contributions of this work and benefits the readers.
- Reviewers question that the proposed framework DAMAS needs the knowledge of the number of environments and the properties of the optimal arms per environment via pretraining. Although the authors claim that they do not assume knowing the true environments and add mismatch experiments where deployed agents don’t match true environments, the design of DAMAS from this perspective is still unclear, both empirically and theoretically.

**Reviewer Scores:**

Reviewer kXPd concerns on the unclear novelty vs. existing works on restless and non-stationary bandits, and the multi-agent aspect seems like nested action space. In addition, the reviewer questions the theory to be not deep. The rebuttal clarifies misunderstandings, but the reviewer’s fundamental concerns on the this is basically UCB with extra machinery and the framework is not novel, may remain.

Reviewer L1tZ questions the mismatch between theoretical and empirical regret, and concerns that the main theory reduces to UCB1-within-phases. The reviewer also have concerns on the empirical studies including the computational overhead, the missing non-stationary baselines, and the robustness to pretraining mismatch. The rebuttal did not fully address those concerns.

Reviewer RXpQ worries about the strong-assumption in the paper. The rebuttal provides mismatch experiments and clarifies the method doesn’t assume knowing true environments or optimal arms. It seems that the reviewer may remain the score.

Reviewer HB9R raises the issues on the asymptotic or weak regret bound and disagreement with "no context" framing.  The rebuttal addresses some of those and the reviewer may remain the score.

---

### Decision · Program_Chairs · 2026-01-26

Reject